# Global Tropopause Altitudes in Radiosondes and Reanalyses

Tao Xian[1,2] and Cameron R. Homeyer[1]

[1]School of Meteorology, University of Oklahoma, Norman, OK 73072, USA
[2]School of Earth and Space Sciences, University of Science and Technology of China, Anhui, 230026, China
**Correspondence:** Cameron R. Homeyer (chomeyer@ou.edu)

**Abstract.** Accurate depictions of the tropopause and its changes are important for studies such as stratosphere-troposphere exchange and climate change. Here, the fidelity of primary lapse-rate tropopause altitudes and double tropopause frequencies in four modern reanalyses (ERA-Interim, JRA-55, MERRA-2, and CFSR) is examined using global radiosonde observations. In addition, long-term trends (1981-2015) in these tropopause properties are diagnosed in both the reanalyses and radiosondes. It is found that reanalyses reproduce observed tropopause altitudes with little bias (typically less than $\pm$ 150 m) and error comparable to the model vertical resolution. All reanalyses underestimate the double tropopause frequency (up to 30 % lower than observed), with the largest biases found in JRA-55 and the smallest in CFSR. The underestimates in double tropopause frequency are primarily attributable to the coarse vertical resolution of the reanalyses. Significant increasing trends in both tropopause altitude (40–120 m per decade) and double tropopause frequency ($\geq$3 % per decade) were found in both the radiosonde observation and the reanalyses over the 35-year analysis period (1981-2015). ERA-Interim, JRA-55, and MERRA-2 broadly reproduce the patterns and signs of observed significant trends, while CFSR is inconsistent with the remaining datasets. Trends were diagnosed in both the native Eulerian coordinate system of the reanalyses (fixed longitude and latitude) and in a coordinate system where latitude is defined relative to the mean latitude of the tropopause break (the discontinuity in tropopause altitude between the tropics and extratropics) in each hemisphere. The tropopause break-relative coordinate facilitates the evaluation of tropopause behavior within the tropical and extratropical reservoirs and revealed significant differences in trend estimates compared to the traditional Eulerian analysis. Notably, increasing tropopause altitude trends were found to be of greater magnitude in tropopause break-relative coordinates and increasing double tropopause frequency trends were found to occur primarily poleward of the tropopause break in each hemisphere.

## 1 Introduction

The tropopause - the boundary between the often unstable, convectively dominated troposphere and stably stratified strato-sphere - is an important boundary for many studies in the atmospheric sciences. For example, long-term changes in tropopause altitude are considered to be an indicator of climate change (e.g., Santer et al., 2003a, b; Añel et al., 2006; Xian and Fu, 2017). Based on radiosonde and satellite observations, previous studies show a significant global rising trend in the tropopause alti-

tude during the last several decades (e.g., Seidel et al., 2001; Santer et al., 2003a; Seidel and Randel, 2006; Feng et al., 2012). Santer et al. (2003a, b) employed output from climate model simulations to assess tropopause trends and attributed the long-term increase to increased greenhouse gases and stratospheric ozone depletion, which leads to a warming of the troposphere and a cooling of the stratosphere. These changes in atmospheric composition enhance the meridional temperature gradient in the upper troposphere and lower stratosphere (UTLS), which in turn strengthens the subtropical jets (to maintain thermal wind balance) and accelerates the Brewer-Dobson circulation (BDC) (e.g. Held, 1993; Kushner et al., 2001; Birner, 2010b; Butchart, 2014; Sioris et al., 2014; Abalos et al., 2015; Fu et al., 2015; Ploeger et al., 2015). The structure and variability of the tropopause also plays a key role in stratosphere-troposphere exchange (STE) studies.The fidelity of the tropopause altitude directly impacts the quantification of STE because the troposphere-stratosphere boundary itself is fundamental to identifying an exchange event and its impact on UTLS composition (e.g., Xian and Fu, 2015; Liu and Liu, 2016; Boothe and Homeyer, 2017). Moreover, the temperature of the tropopause, especially in the tropics, regulates the transport of water vapor (a powerful greenhouse gas) from the troposphere to the stratosphere (Holton et al., 1995; Fueglistaler et al., 2009).

Several tropopause criteria have been used in prior studies and are based on thermal, dynamical, and chemical characteristics of the atmosphere (WMO, 1957; Holton et al., 1995; Kunz et al., 2011, 2015; Pan et al., 2004). The original tropopause definition is based on the temperature lapse rate (the negative of the vertical temperature gradient) according to criteria put forth by the World Meteorological Organization (WMO, 1957). This lapse-rate tropopause is globally reliable and found to commonly coincide with the sharpest stability and chemical transitions between the troposphere and stratosphere (Pan et al., 2004; Gettelman et al., 2011). The only known exceptions to this reliability of the lapse-rate tropopause occur within complex UTLS stability environments (e.g., layered UT and LS air near the subtropical jet during Rossby wave breaking events; Homeyer et al., 2011) and over the Antarctic during austral winter, where there exists an erroneously high lapse-rate tropopause altitude due to weakened stability in the lower stratosphere (Zängl and Hoinka, 2001). The issue over the Antarctic is confined to latitudes poleward of $60°S$ for about 3 months out of the year. The WMO definition also allows for identification of more than one tropopause if low stability layers are observed for a substantial depth above the primary tropopause. Double lapse-rate tropopauses have been the focus of many studies during the past two decades and have been found to be largely related to Rossby wave breaking events and associated STE above the subtropical jets (e.g., Shapiro, 1980; Seidel and Randel, 2006; Pan et al., 2009; Homeyer et al., 2011; Añel et al., 2012; Peevey et al., 2014; Schwartz et al., 2015; Manney et al., 2017; Liu and Barnes, 2018).

Tropopause definitions based on either the maximum static stability gradient in the vertical dimension or curve fitting to the static stability profile, which is typically characterized as a step function from uniformly low stability in the troposphere and high stability in the stratosphere, have been increasingly used in global tropopause studies (Birner, 2010a; Homeyer et al., 2010; Gettelman and Wang, 2015). These approaches often provide unique information on tropopause structure, such as its sharpness (the depth of the troposphere-stratosphere stability transition). However, static stability definitions frequently fail in the subtropics where the stability profile is often layered (e.g., double lapse-rate tropopauses), with maxima at multiple altitudes or at altitudes well removed from the most prominent transition, or the transition from troposphere to stratosphere occurs over a deep ($\geq$3 km) layer.

Alternative definitions to the lapse-rate tropopause or one based on static stability are often applied in specific locations, separated mostly by latitude. In the tropics, the UTLS temperature minimum, known as the cold point, is often used to define the tropopause. This cold point tropopause is typically employed in studies that examine troposphere-to-stratosphere transport of water vapor (Holton et al., 1995; Mote et al., 1996; Fueglistaler et al., 2009). While easily defined, cold point tropopause

altitudes are only reliable within the deep tropics (between 20°S and 20°N), because outside of this region the coldest temperature in a vertical profile is not always associated with the transition between tropospheric and stratospheric air (indicated by stability or composition). Profiles containing multiple lapse-rate tropopauses are a good example of when the cold point tropopause fails. Potential vorticity (PV), which is conserved in an adiabatic and friction-less flow, is commonly used for transport studies in the extratropics and often used to define a dynamical tropopause (Holton et al., 1995; Kunz et al., 2011, 2015;

Homeyer and Bowman, 2013; Boothe and Homeyer, 2017). The PV threshold used ranges from $\pm 1$–4 PVU (where 1 PVU $= 10^{-6}$ km$^2$ kg$^{-1}$ s$^{-1}$) in previous studies, with most using $\pm 2$ PVU (e.g. Wernli and Bourqui, 2002; Sprenger et al., 2003; Sprenger and Wernli, 2003). The PV value that best coincides with the lapse-rate tropopause varies with latitude and season and, if this variability is not accounted for, it can result in large differences in quantitative transport studies (Hoinka, 1998; Homeyer and Bowman, 2013). For example, Homeyer and Bowman (2013) found that changing the PV iso-surface from $\pm 2$ to

$\pm 4$ PVU resulted in a reversal of the net STE between the tropics and extratropics from Rossby wave breaking. The dynamical tropopause is only reliable in close proximity to and poleward of the subtropical jets, since PV approaches a value of 0 near the equator and iso-surfaces diverge from the lapse-rate and cold point tropopauses in the deep tropics.

One final type of tropopause definition that has been used in many studies is that based on chemical composition. Multiple studies have used ozone ($O_3$) profiles to define the tropopause, where the $O_3$ tropopause is defined using absolute thresholds

for $O_3$ concentration and vertical gradients of $O_3$ (Bethan et al., 1996; Wild, 2007). Unique limitations exist for $O_3$ tropopause altitudes due to the seasonality and location-dependent variability of the ideal $O_3$ concentration threshold value (Logan, 1999). Another chemical tropopause definition exists that uses multiple coincident trace gas concentrations to define the UTLS chemical transition layer, often leveraging $O_3$ and carbon monoxide (Zahn et al., 2004; Pan et al., 2004). However, such coincident chemical observations are uncommon, making the method impractical for climatological analysis. Artificial tracers have be-

come increasingly used in numerical models to allow for a chemical tropopause definition, with the 90-day lifetime tracer (known as e90) being a widely used choice in the chemistry climate model community (Prather et al., 2011). Primary limitations of this approach are that it cannot be applied to observations and it is not incorporated into reanalysis models, which are commonly used for climatological analyses.

In summary, multiple tropopause definitions exist in the community and are often chosen based on the goals of the study.

The primary goal of this study is to evaluate long-term changes in tropopause characteristics globally. Since the lapse-rate tropopause is a global definition that can be easily applied to both conventional observations and model output, agrees with the sharp stability and chemical transitions between troposphere and stratosphere, and enables the unique opportunity to study multiple tropopause structures, we employ this definition for analysis in this study.

Radiosondes have been the traditional source of thermodynamic profiles of the atmosphere from the near surface up to 30

km since 1950s, and have been widely used for studying long-term changes in the tropopause (e.g. Seidel and Randel, 2006;

Añel et al., 2007, 2008; Xian and Fu, 2015). A primary shortcoming of radiosonde observations is the limited spatial coverage, since they are mostly launched from land masses. Another limitation is that not all radiosonde flights from a given location are successful, leading to discontinuities in the data record. Moreover, despite the fact that the number of sites providing operational radiosonde observations has increased over time, the number of locations with long-term records suitable for trend studies is relatively small. More recently, modern high-resolution reanalysis models have been used to evaluate tropopause characteristics globally since they provide data that is spatially and temporally continuous (Manney et al., 2014, 2017; Boothe and Homeyer, 2017). Reanalyses assimilate global quality-controlled observations to provide best estimates of past three-dimensional atmospheric states and are used to develop an understanding of a wide range of atmospheric processes, which is often not possible using observations alone (Fujiwara et al., 2017). Several reanalyses are publicly available and cover historical periods of 30 years or longer, with modern reanalyses such as MERRA and MERRA-2 (the National Aeronautics and Space Administration Modern-Era Retrospective analysis for Research and Applications, Versions 1 and 2), ERA-Interim (the European Centre for Medium-Range Weather Forecasts interim reanalysis), JRA-55 (the Japanese Meteorological Association 55-year reanalysis), and CFSR (the National Centers for Environmental Prediction Climate Forecast System Reanalysis) being widely used for research today.

While tropopause altitudes can be similarly evaluated in observations and reanalyses, the reanalyses can provide us with a broader understanding of tropopause behavior given the spatial and temporal limitations of the observations outlined above. For example, Manney et al. (2017) compared the frequency of double tropopauses in five modern reanalyses during 1980–2014 and highlighted the sensitivity of the double tropopause identifications to model vertical resolution. Boothe and Homeyer (2017) further argued that differences in STE estimates based on reanalysis output are partly due to the differences in vertical grid spacing. Moreover, reanalyses are highly dependent on the underlying global forecast models, data input sources, and assimilation systems (Fujiwara et al., 2017). For instance, although both ERA-Interim and MERRA-2 assimilate $O_3$ profiles from the Aura Microwave Limb Sounder, they use climatological and prognostic $O_3$ fields for radiation calculations, respectively (Dethof and Hólm, 2004; Rienecker et al., 2008). This, in turn, may have a significant impact on assimilated ozone and stratospheric temperatures during winter and spring, which could impact tropopause calculations.

Here, we investigate the accuracy of primary tropopause altitudes and double tropopause frequency in four modern reanalyses (ERA-Interim, JRA-55, MERRA-2, and CFSR) and diagnose long-term (1981-2015) trends in tropopause altitude and double tropopause frequency using both radiosonde observations and reanalyses. In particular, we address three research questions: (1) How well do modern reanalyses represent the lapse-rate tropopause? (2) What are the recent trends in tropopause altitude and double tropopause frequency and how do they vary spatially?, and (3) How sensitive are tropopause altitude trends to the geographic coordinate system used? This research provides a unique evaluation of model performance, of climate variability and change, and a physical perspective of UTLS dynamics, including STE that is commonly associated with double tropopause events.

## 2    Data and Methods

### 2.1    Reanalysis output

ERA-Interim output is available from 1979 to the present on an approximately 80 km horizontal grid and with 750-1250 m vertical resolution in the UTLS (Dee et al., 2011), where the UTLS is defined here as the 8-18 km altitude layer. JRA-55 is

available from 1958 to the present on a ∼60 km horizontal grid (Kobayashi et al., 2015). Similar to ERA-Interim, JRA-55 has 750-1250 m vertical grid spacing in the UTLS. MERRA-2 is available from 1979 to the present on a $0.5° \times 0.625°$ longitude-latitude grid and at ∼1100 m vertical resolution throughout the UTLS (Bosilovich, 2015). CFSR is available from 1979 to 2010 on a $0.5° \times 0.5°$ longitude-latitude grid and at 700-900 m vertical resolution in the UTLS. CFSR output is extended to the year 2015 in this study using analyses from the Climate Forecast System version 2 (CFSv2) model. For a more detailed

discussion of these reanalyses and their differences (including profiles of vertical resolution), see Fujiwara et al. (2017). All tropopause analyses are done using daily 0000 UTC fields from each reanalysis (though not shown, analyses using alternative synoptic times for shorter time periods are consistent). Geopotential height was computed for each reanalysis model-level output using the moisture-included hypsometric equation. Meteorological parameters are interpolated linearly to a regular $1° \times 1°$ longitude-latitude grid in the horizontal for analysis to enable 1-to-1 comparison (this choice has a negligible impact on

the reported results). Temperature profiles are linearly interpolated to a regular 200-m vertical resolution prior to tropopause identification.

### 2.2    Global radiosonde data

Radiosonde data used in this study were obtained from the Integrated Global Radiosonde Archive (IGRA) Version 2 (Durre et al., 2016). The IGRA database provides historical radiosondes from locations around the world that have been comprehen-

sively quality controlled and corrected for gross errors. These data are the best long-term historical record available for studies of the vertical structure of the UTLS (global satellite-based observations, such as radio occultation, are available for only the last ∼18 years). IGRA radiosonde observations are mainly available twice daily at 0000 and 1200 UTC. Given that a small number of stations launch radiosondes at non-standard times, we included launches that occurred between 2100 and 0300 UTC in the 0000 UTC analysis, and those between 0900 and 1500 UTC in the 1200 UTC analysis. The IGRA radiosondes are

provided at mandatory (conventional pressure levels) and significant levels, which preserves any substantial lapse-rate changes in the original data (observations are typically taken at 6-s intervals, resulting in ≤50-m vertical resolution), but can result in coarse vertical resolution in the UTLS (>1 km). Such coarse resolution of the profile can prevent successful application of the lapse-rate tropopause by limiting the number of observations used to satisfy the second criterion of the WMO definition and the criterion for identifying multiple tropopauses (see Section 2.3 below), as is true for the reanalysis output. Thus, in order

to enable thorough evaluation of the WMO criteria and reliable tropopause identification, radiosonde data are also linearly interpolated to a 200-m regular vertical grid prior to tropopause identification. Only soundings that have valid observations between 5 km and 22 km altitude are used to identify the tropopause. The original high vertical resolution (∼5 m) profiles from select National Weather Service sites in the United States were also retrieved for illustration purposes only (see Figure 1).

## 2.3  Lapse-rate tropopause identification

As outlined in the Introduction, we employ the WMO lapse-rate tropopause definition for analysis in this study. The WMO definition defines the first tropopause in a profile as "the lowest level at which the lapse rate falls to 2 °C km$^{-1}$ or less, provided also the average lapse rate between this level and all higher levels within 2 km does not exceed 2 °C km$^{-1}$" (WMO, 1957).

The WMO definition allows for a secondary tropopause "if above the first tropopause the average lapse rate between any level and all higher levels within 1 km exceeds 3 °C km$^{-1}$, then a second tropopause is defined by the same criterion." To avoid boundary layer inversions and false tropopause identification, as well as secondary tropopauses above the altitude of that typically observed for the primary tropopause in the tropics, the algorithm is applied only to altitudes ranging from 5 km to 22 km. Lapse rates are calculated for each profile using a forward (upward) difference scheme in the form $\Gamma(z_i) = -\partial T / \partial z \approx$

$-(T_{i+1} - T_i)/(z_{i+1} - z_i)$, where $\Gamma$ is the lapse rate in °C km$^{-1}$, $T$ is temperature in °C, and $z$ is altitude in km. Tropopause altitudes from reanalysis are geopotential altitudes.

Example WMO lapse-rate tropopause altitudes calculated for two randomly selected radiosonde profiles launched at the Corpus Christi, Texas National Weather Service office (97.5°W longitude, 27.78°N latitude) in the United States are shown in Figure 1. Both of these profiles have two WMO tropopauses. Here, we show both the full resolution temperature profile ob-

tained by each radiosonde and the reduced resolution profile from the IGRA archive to demonstrate common differences in the level of detail between native data and that reported at mandatory and significant levels. As demonstrated in both profiles, the IGRA data preserves all significant lapse rate transitions and results in nearly equivalent tropopause altitude definitions to those computed using the full resolution profile (differences are ≤100 m). Consistent results are found when selecting profiles randomly from other stations and time periods (not shown). Thus, we are confident that mandatory and significant levels included

in the IGRA data are suitable for tropopause analyses following the methods employed here. Coincident temperature profiles and tropopause altitudes from each reanalysis model are superimposed in Figure 1 and show much less detail than the higher resolution observations and a general inability to capture multiple tropopause altitudes. These example profiles have shallow inversion layers above the primary lapse-rate tropopause altitude, which are often not well captured in the coarse resolution reanalysis model profiles. As a result, the secondary tropopause is often erroneously classified as the primary tropopause altitude

in model output (e.g., see also Figs. 4 & 6 and discussion from Homeyer et al., 2010). Extensive comparisons of tropopause altitudes and multiple tropopause frequencies between the radiosondes and reanalyses are provided in Section 3.1.

## 2.4  Trend analyses

Trends over the 35-year analysis period (1981-2015) of this study are calculated using monthly mean primary tropopause altitudes and the monthly fraction of profiles with double tropopauses. For radiosondes, we require a station to have at least 20

days of suitable profiles (all from either 00 UTC or 12 UTC and including mandatory and significant levels up to an altitude of 22 km or higher) to compute a monthly mean tropopause altitude or double tropopause fraction. Trends are only calculated for radiosonde stations that have sufficient observations for at least half of the total number of months in the 35-year period and a roughly even distribution of data points throughout the period (data gaps < 5 years and no missing data near the beginning and

end of the time series) for adequate trend analysis. Monthly tropopause time series from both radiosondes and reanalyses are then deseasonalized using a high-pass filter that removes variability at time scales less than or equal to 1 year. Linear regression is used on the filtered time series to measure trends over the 35-year period. Trends are deemed significant if they exceed the 3-$\sigma$ uncertainty of the measured slope, which is analogous to statistical significance at the 99% confidence level for a Gaussian distribution.

For reanalyses, trends are also calculated in an alternative coordinate system. A sharp discontinuity in the primary lapse-rate tropopause altitude is found near the subtropical jet and known as the "tropopause break" (e.g., Randel et al., 2007; Pan and Munchak, 2011; Homeyer and Bowman, 2013). Tropopause altitudes are uniformly high ($\geq$15 km) in the tropics and uniformly low (mostly 8–12 km altitude) in the extratropics. In fact, as shown by Birner (2010b), Boothe and Homeyer (2017) and others, global and hemispheric frequency distributions of tropopause altitude are bimodal. As a result, the tropopause break is easily defined in model output as globe circling contours of a threshold tropopause altitude or pressure coinciding with the frequency minimum between the tropical and extratropical modes (typically $\sim$14 km or $\sim$150 hPa; e.g., see Homeyer and Bowman, 2013). Because the location (latitude) of the subtropical jets and tropopause breaks varies considerably in space and time, trend analyses in the vicinity of the tropopause breaks can be adversely impacted by their variability and potential long-term changes in latitude. Thus, to remove this variability and evaluate trends within the tropical and extratropical reservoirs separately, we also analyze trends in the reanalyses using a tropopause break-relative latitude coordinate. Tropopause break latitudes are identified at each 00 UTC analysis using contours of the tropopause altitude that coincides with the frequency minimum between tropical and extratropical modes in each hemisphere. Monthly means are then calculated by averaging the instantaneous tropopause fields on the relative latitude grid in each hemisphere. For plotting, tropopause break-relative analyses are mapped using the long-term mean break latitudes and any data extending beyond the equator and pole is trimmed from each hemisphere.

Note that time series used for trend analysis were not adjusted for potential discontinuities and/or biases owing to changes in instrumentation or other factors (e.g., see Añel et al., 2006; Antuña et al., 2009). Thus, some elements of the trend analyses (especially for the radiosonde observations) may be impacted by these artifacts. However, a substantial number of time series with large, significant trends (see Section 3.2) were manually evaluated and no discontinuities were found (not shown). Thus, we expect such factors to have a minimal impact on the results outlined below.

## 3 Results

### 3.1 Tropopause validation

To evaluate the fidelity of tropopause altitudes in the reanalyses, we first compare tropopause altitudes from the gridded reanalysis output with the radiosonde data at 00 UTC only. Instantaneous tropopause altitudes computed on the 1°$\times$ 1° longitude-latitude grids are interpolated linearly in space to the locations of the radiosondes for comparison. Results of these comparisons are shown for individual months during each season: January, April, July and October between 1981 and 2015. In total, 317 radiosonde stations and approximately $9.9\times10^4$ profiles are used for this validation, with their geographic locations shown as

circles in Figure 4. Biases and errors (r.m.s. differences) in reanalysis primary tropopause altitudes from this comparison are listed in Table 1. More than half of the primary tropopause altitudes in MERRA-2 are found to have a positive bias, whereas the majority of biases in the remaining reanalyses (ERA-Interim, JRA-55, and CFSR) are negative. Errors in the tropopause altitudes range from 950 m to 1200 m, which is comparable to the vertical grid spacing of the reanalyses.

All reanalyses produce too few double tropopauses compared to the radiosonde observations. Double tropopause frequency in the reanalyses is significantly underestimated, with 67-85 % of the sample having a negative bias. Biases are largest in JRA-55, with over 22 percentage points fewer double tropopauses than observed, and smallest in CFSR, which has 13 percentage points fewer double tropopauses than observed. Errors in double tropopause frequency show similar differences among the reanalyses. The lower double tropopause frequency in reanalyses is likely due to the coarse vertical resolution of the mod-
els. As outlined in Section 2.3 and illustrated in Figure 1, accurate tropopause identification requires vertical resolution of ≤1 km to detect shallow low and high stability layers that are often responsible for the occurrence of multiple tropopauses. CFSR, which has the lowest bias and error also has the finest vertical grid resolution in the UTLS, while MERRA-2 typically has finer resolution than ERA-Interim and JRA-55 at the altitude of the secondary tropopause. The differences in bias and r.m.s. error between models with approximately equivalent vertical grids, such as ERA-Interim and JRA-55, suggest that the
representation of atmospheric processes and/or the data assimilation system used could also be responsible for some of the under-representation of double tropopause events.

      To better evaluate the role of model vertical resolution as a source for the observed tropopause differences, we repeated the validation with degraded radiosonde observations. In particular, each IGRA radiosonde profile was linearly interpolated to the fixed model-level grid of each reanalysis in order to limit the detail of the observed temperature profile to that available from
each model. These degraded radiosonde profiles were then used to calculate unique observation-based tropopause altitudes to compare with each reanalysis and determine the resulting bias and error, which is expected to be reduced (especially for double tropopauses). Table 1 shows these evaluations using the degraded radiosonde profiles and confirms that the biases and errors in both primary tropopause altitude and double tropopause frequency largely decrease. However, some bias and error remains (and for MERRA-2 tropopause altitudes, increases), which suggests that alternative sources of error (e.g., data assimilation,
model physics/dynamics) are significant.

      In addition to overall performance of the reanalyses, there are seasonal regional variations in the tropopause biases and errors. In order to better understand the source of these variations, we group the comparisons by latitude to reduce potential sources of uncertainty from the non-uniform global distribution of the radiosonde locations. Figures 2 and 3 show the bias and r.m.s. error for primary tropopause altitudes and double tropopause frequency, respectively, within five latitude bands:
two in the extratropics of each hemisphere (45–90°N and 45–90°S), two in the subtropics of each hemisphere (20–45°N and 20–45°S), and one in the deep tropics (20°S–20°N). The southern hemisphere subtropics and extratropics tend to have larger errors than their counterparts in the northern hemisphere, likely due to (in part) fewer data sources for assimilation and fewer profiles used for analysis. Regionally, the largest primary tropopause errors are found within the subtropics of each hemisphere - commonly associated with the location of the subtropical jet and tropopause break. The complicated stability structure within
the subtropics is known to lead to large errors in tropopause altitudes within models and is primarily the result of inadequate

representation of multiple tropopauses and the precise location of the tropopause break (Homeyer et al., 2010). The largest differences within the subtropics occur during the winter season of each hemisphere, which is consistent with the time period during which the subtropical jet reaches its maximum intensity (Manney et al., 2017) and double tropoapuses are more frequent (e.g. Randel et al., 2007; Añel et al., 2008; Manney et al., 2017). The tropopause break is also sharpest in the vicinity of the

subtropical jets at this time due to thermal wind balance (i.e., the latitudinal temperature gradients in the vicinity of the jet are largest). Biases and errors in double tropopause frequency are largest in the subtropics and extratropics of each hemisphere and small within the tropics, where multiple tropopauses are generally infrequent. Biases and errors are also largest in the winter of each hemisphere and smallest during the summer.

## 3.2    Eulerian mean tropopause altitude trends

The long-term trend in the tropopause altitude is considered to be an indicator of climate change. Here, we evaluate trends in Eulerian monthly mean primary tropopause altitudes using both radiosondes and reanalyses (Figure 4). Statistically significant trends in tropopause altitude are found in the radiosonde observations for many locations across the globe. Most trends point to an increase in tropopause altitude over time, with the largest increases (100-200 m per decade) found over western China, the contiguous United States, eastern Europe, and Indonesia. A small region of significant decreasing primary tropopause altitudes

is found over Siberia and the subtropical Pacific.

Significant tropopause altitude trends in the reanalyses are in broad agreement with those identified from the radiosonde observations. In particular, regions with dense radiosonde coverage and large, positive trends are well represented in each reanalysis. The small, negative trend identified over Siberia is also reproduced in each reanalysis and agrees with previous studies (Santer et al., 2003a, b; Añel et al., 2006), but this behavior may be an artifact of changes in instrumentation and

quality control of the radiosonde data over time (Añel et al., 2006). Trends in the reanalyses are generally larger and more variable outside of the regions with dense radiosonde coverage. Some notable features are the upward/downward tropopause altitude trend dipoles found over the eastern subtropical Pacific in ERA-Interim, JRA-55, and MERRA-2. This dipole is not found in CFSR and the magnitudes of the trends and spatial extent of the dipole vary considerably in the remaining reanalyses. Moreover, the dipole is consistent with significant narrowing of the tropics over the eastern Pacific that has been identified in

the reanalyses via subtropical jet and tropopause break analysis (Manney and Hegglin, 2018; Martin et al., 2018).

Another notable difference among the reanalyses is the depiction of tropopause trends across the tropics, where MERRA-2 and CFSR show considerably larger upward trends than ERA-Interim and JRA-55. ERA-Interim also depicts a downward trend over the central Pacific, which is not observed in the remaining reanalyses. Trends in JRA-55 and MERRA-2 appear to be more consistent with the limited number of radiosondes available in the tropics, especially over Indonesia, the central

Pacific, and Northern Australia. Finally, the tropopause altitude trends over Antarctica are found to be inconsistent among the reanalyses. Namely, a rising trend is found over Antarctica in ERA-Interim and MERRA-2, while a less extensive decreasing trend is found in JRA-55 and CFSR.

### 3.3 Break-relative tropopause altitude trends

As discussed briefly in the previous section, recent studies have identified significant regional changes in the width of the tropics that introduce some uncertainty to the precise nature of tropopause changes when diagnosing trends in an Eulerian framework. Therefore, in order to mitigate the effects of a meandering tropopause break and focus on tropopause changes in the tropical and extratropical reservoirs alone, we employ a tropopause break-relative coordinate here. Figure 5 shows the geographic distribution of primary tropopause altitude trends in a tropopause break-relative altitude coordinate for each reanalysis from 1981 to 2015. Considerable differences are found in the diagnosed trends here compared to those using the Eulerian monthly mean fields. In particular, significant trends are larger in magnitude in the tropopause break-relative coordinate and are more consistent amongst the reanalyses, especially over the eastern subtropical Pacific. ERA-Interim, JRA-55, and MERRA-2 are broadly consistent with one another, with large upward trends in tropopause altitude in the extratropics over the Pacific and similar patterns and signs of significant trends elsewhere.

MERRA-2 shows substantially different trends throughout the tropics relative to ERA-Interim and JRA-55, with both larger upward magnitudes and unique patterns over the western Pacific and Indian Ocean. CFSR shows some consistency with MERRA-2 in the tropics in terms of the trend magnitude, but is inconsistent in pattern (especially over the Americas). In ERA-Interim, JRA-55, and MERRA-2, trends in the tropics are largest immediately equatorward of the tropopause break latitude in each hemisphere, while CFSR shows the greatest trends in the deep tropics. However, the Eulerian mean trends in Figure 4 show more consistent behaviour amongst the reanalyses in the depiction of these poleward maxima in altitude trends within the tropics, which is in agreement with previous analyses (e.g. Seidel and Randel, 2006). Outside of the tropics, CFSR is in broad disagreement with the remaining reanalyses and suggests largely decreasing trends in tropopause altitude.

### 3.4 Double tropopause climatology and trends

As outlined in Section 3.1, double tropopause frequencies often have large bias and error in the reanalyses, which is largely attributed to their coarse vertical grid resolution. To better understand the nature of these errors, maps of annual-mean double tropopause frequency are presented for the radiosondes and reanalyses in Figure 6. These maps are consistent with similar analyses of radiosondes, satellite observations, and reanalyses in prior studies (e.g., Randel et al., 2007; Añel et al., 2008; Peevey et al., 2012; Manney et al., 2017), and show that there are belts of high double tropopause frequency in the northern subtropics and midlatitudes of each hemisphere, largely near and poleward of the subtropical jets and tropopause breaks. The patterns and spatial extent of these belts are consistent between the radiosonde observations and reanalyses, but there are considerable differences in the frequency values. Radiosondes show that these high-frequency belts are characterized by values ≥40 %, while the reanalyses are at least 10–20 % lower. The magnitudes of the differences between the radiosonde and reanalysis frequencies within the high-frequency belts are consistent with that found in the overall bias and error evaluation (Fig. 3). Outside of the high-frequency belts, there is a unique feature found within the tropics in CFSR. Namely, a narrow band of double tropopause frequency between 10 and 30 % is seen along the equator stretching from central Africa to eastern Indonesia. This feature does not exist in the remaining analyses and is poorly sampled by the radiosonde network. However,

the small number of stations available in western Indonesia do show consistent double tropopause frequencies. These double tropopauses may be driven (in part) by shallow, lateral transport of extratropical lower stratospheric air into the tropical upper troposphere on the eastern edge of the Asian monsoon anticyclone (e.g., Konopka et al., 2010), but the lack of continuity of this feature between the midlatitude high-frequency belt and the enhanced frequency belt along the equator suggests that the dynamics of the monsoon anticyclone may also be important to their formation.

Eulerian mean trend maps for double tropopause frequency during 1981-2015 are shown for the radiosondes and reanalyses in Figure 7. Trends in the double tropopause frequency are found to be statistically significant at almost all of the radiosonde locations and show substantial increases in frequency ($\geq 2$ % per decade) nearly everywhere. The largest increasing trends ($\geq 3$ % per decade) for double tropopause frequency are found in the midlatitudes in each hemisphere and poleward of the high-frequency belts in the long-term climatology, with some small (mostly $<1$ % per decade) decreasing trends over Siberia, southern China and the Caribbean (locations with climatologically low double tropopause frequency). The midlatitude increasing trends are comparable to those diagnosed in Castanheira et al. (2009) between 1970 and 2006 for the 30–60°N and 30–60°S latitude belts, which were 3.3 % and 6.6 % per decade, respectively. Taken together with the climatological double tropopause distribution, these trends imply that the area of frequent double tropopause environments is increasing in each hemisphere, mostly indicating a northward expansion of the high-frequency belts.

Areas of significant increasing trends in double tropopause frequency are largely consistent between the radiosondes and reanalyses, with CFSR being the only exception. In particular, ERA-Interim, JRA-55, and MERRA-2 all show large increases in double tropopause frequency along and mostly north of the tropopause break. CFSR shows mostly decreasing trends in double tropopause frequency across the globe and is broadly inconsistent with the radiosonde observations. Where areas of significant trends agree in pattern and sign between the reanalyses and radiosondes, the magnitudes are largely consistent near the tropopause break and inconsistent poleward of the break. ERA-Interim, JRA-55, and MERRA-2 do not reproduce well the poleward extent of the significant increasing trends in observations, especially over North America. A unique increasing trend is found along the equator in MERRA-2, which is consistent with the location of the narrow band of moderate double tropopause frequency found in the CFSR climatology. While none of the remaining reanalyses show this feature, it is consistent with trends observed in the small number of radiosonde stations over western Indonesia and the western Pacific. Finally, the double tropopause frequency trends over Antarctica are decreasing in all reanalyses, but the area and magnitude of the trend varies considerably. The most consistent element of this feature is found over western Antarctica, but there are no radiosonde observations in this region to validate such a trend and the climatological frequencies in this region are small.

Maps of annual-mean double tropopause frequency and frequency trends in tropopause break-relative coordinates from the reanalyses are shown in Figures 8 and 9, respectively. In general, there is less variation in the patterns of both high-frequency double tropopause regions and increasing double tropopause frequency trends between the Eulerian and tropopause break-relative analyses. Annual-mean frequencies are higher in tropopause break-relative coordinates and maximize poleward of the mean tropopause break latitude throughout each hemisphere. Significant increasing double tropopause frequency trends are also found primarily poleward of the tropopause breaks in each hemisphere, but are otherwise mostly consistent with the Eulerian analysis. Two notable exceptions are the areas of significant increasing trends over the Northern and Southern east

Pacific and east Atlantic in ERA-Interim, JRA-55 and MERRA-2, where the largest increasing trends were found to be mostly equatorward of the mean tropopause break latitude in the Eulerian analysis. In tropopause break-relative coordinates, the areas of greatest increasing trends are generally centered on or poleward of the tropopause break in the Northern Hemisphere and poleward of the tropopause break in Southern Hemisphere.

## 4 Conclusions and discussion

In this study, we examined the fidelity of primary tropopause altitudes and double tropopause frequency in four modern reanalyses (ERA-Interim, JRA-55, MERRA-2, and CFSR) using the WMO lapse-rate tropopause definition. Long-term trends in the primary tropopause altitude and double tropopause frequency over a 35-year period (1981-2015) were also examined using both radiosonde observations and reanalyses. All reanalyses were found to reproduce observed primary tropopause altitudes with little bias and error comparable to the vertical grid resolution of the models, which is consistent with previous model tropopause evaluations (e.g., Homeyer et al., 2010; Solomon et al., 2016a). Bias and errors in the primary tropopause altitude were found to vary regionally, with the largest magnitudes of both routinely found in the subtropics of each hemisphere (Fig. 2). Double tropopause frequencies are broadly underrepresented in the reanalyses, with biases of up to 30 percentage points lower than observed. JRA-55 consistently showed the largest double tropopause bias, while CFSR consistently showed the lowest. The majority of error in double tropopause frequency was found in the subtropics and high latitudes of each hemisphere, where double tropopause environments are most common. Based on the differences in vertical grid resolution of the models and the necessary conditions for multiple tropopause identification using the WMO definition, the underestimates in double tropopause frequency in the reanalyses are argued to primarily be the result of too coarse vertical grid spacing. When the radiosonde data are degraded to the vertical grid resolution of each reanalysis, the biases and errors in double tropopause frequency are greatly reduced, while the biases and errors in primary tropopause altitude show little sensitivity to this change (Table 1; Figs. 2 & 3).

Trends in primary tropopause altitudes were found to be significant and increasing (i.e., upward) across most of the globe in the radiosonde observations, largely ranging from 40 to 120 m per decade (Fig. 4). Some similar, but significant decreasing altitude trends were found for a few radiosonde stations in Siberia. The reanalyses broadly reproduce the patterns and signs of significant trends in the radiosonde observations, with some disagreement in the areas of significant upward trends over China, Australia, and northern Antarctica. Outside of the regions with dense radiosonde coverage, there are significant upward and downward primary tropopause altitude trend dipoles over the central and eastern subtropical Pacific in ERA-Interim, JRA-55, and MERRA-2. Trend patterns in CFSR in this region are not consistent with the remaining reanalyses. To limit the impact of frequent meandering of the tropopause break and long-term trends in its location to the diagnosed primary tropopause altitude trends within the tropical and extratropical reservoirs, we also computed trends in a tropopause break-relative latitude coordinate (Fig. 5). The break-relative analysis revealed larger trends ($\geq$120 m per decade) in both the tropics and extratropics, which were increasing nearly everywhere and greatest within the tropics immediately equatorward of the tropical break latitudes in each hemisphere and in the extratropical reservoir over the eastern Pacific. As found in the Eulerian tropopause trend analysis, ERA-Interim, JRA-55, and MERRA-2 showed consistent patterns and magnitudes of significant

trends, except for some locations within the tropics. CFSR was once again broadly inconsistent with the remaining reanalyses and showed decreasing tropopause altitude trends throughout most of the extratropical reservoir.

Depending on the reference frame (Eulerian or tropopause break-relative), the maxima in tropopause altitude trends in the tropics immediately equatorward of the mean tropopause break latitudes may indicate widening of the tropics and/or changes in the strength of the subtropical jets. Changes in jet speed can impact tropopause altitudes through changes in the magnitude of the associated vertical ageostrophic circulations around the jets. These circulations advect tropical upper troposphere air poleward above the jet altitude and extratropical lower stratosphere air downward and equatorward below in regions where the jet speed is increasing from west to east. Thus, changes in tropopause altitude near the tropopause break latitudes can be dynamically forced, with lower extratropical tropopause altitudes poleward of the jet and higher tropopause altitudes equatorward of the jet in regions where the west-to-east gradient in jet wind speed is increasing and vice versa in regions where the west-to-east gradient is decreasing. Manney and Hegglin (2018) find decreasing trends in subtropical jet wind speeds in the eastern Pacific within the Northern Hemisphere and increasing trends elsewhere, and increasing trends in subtropical jet wind speeds in the eastern Pacific and decreasing trends over the Indian Ocean within the Southern Hemisphere (e.g., see their Figure 8). Thus, dynamically driven changes in tropopause altitude near the subtropical jet are expected to be upward within the tropics from the eastern Pacific across North America and decreasing from Asia across the central Pacific in the Northern Hemisphere, while dynamically driven trends are expected to be smaller in the Southern Hemisphere. There are some patterns tropopause altitude trends that are consistent with this expectation, but it does not appear to be a major source of the diagnosed trends.

Patterns in tropopause altitude are also expected to be driven (in part) by the geographic distribution of observed surface (and tropospheric) warming during the 1981-2015 period. Figure 10 shows changes in global surface temperatures during the 35-year analysis period from the NASA Goddard Institute for Space Studies (GISS) surface temperature analysis (GISTEMP Team, 2018; Hansen et al., 2010). Patterns of long-term surface warming are consistent with the diagnosed trends in tropopause altitude here, particularly within the midlatitudes. For example, the two prominent regions of tropopause altitude increases found in the eastern midlatitude Pacific within each hemisphere coincide with locally enhanced surface warming during this time period. In addition, the patterns of increasing tropopause altitude trends over North America and Greenland also closely resemble patterns in surface warming there. Decreasing trends in the mid-to-high latitudes of the Southern Hemisphere found in the reanalyses are also consistent with decreasing trends in surface temperatures. These similarities imply that surface (and tropospheric) warming/cooling may be a significant source of diagnosed tropopause altitude trends in the extratropics. Sources of the upward trends within the tropics and their patterns are less clear.

Primary tropopause altitude trends over Antarctica were found to vary considerably among the reanalyses, with increasing trends in ERA-Interim and MERRA-2 and decreasing trends in JRA-55 and CFSR. These conflicting trends over Antarctica may be a result of different ozone input sources and assimilation as well as the dynamical responses to ozone concentration changes (e.g., Martineau et al., 2016; Polavarapu and Pulido, 2017; Fujiwara et al., 2017). For example, ERA-Interim and MERRA-2 assimilate ozone retrievals (both profiles and total column ozone (TCO)) from SBUV and SBUV/2, as well as TCO from OMI and profiles from Aura MLS, while JRA-55 only assimilates TCO and CFSR assimilates TCO and relatively

coarse vertical resolution profiles from SBUV and SBUV/2. In particular, the rising Antarctic tropopause for ERA-Interim and MERRA-2 may be associated with a strengthening of the stratospheric polar vortex, which results in an elevated tropopause altitude due to the anomalous residual meridional upwelling in the polar latitudes and downwelling in the midlatitudes (Kidston et al., 2015). The decreasing tropopause altitudes in JRA-55 and CFSR may be associated with stratospheric ozone recovery (Son et al., 2009; Solomon et al., 2016b) and an acceleration of the BDC. Future work is needed to better elucidate the contributions from these known processes to the diagnosed long-term tropopause altitude trends. If possible, additional research on tropopause characteristics and variability using observations in this region would also be helpful.

Significant increasing trends in double tropopause frequency were found nearly everywhere in the radiosonde observations, with the largest trends near and poleward of the tropopause break ($\geq 3$ % per decade; Fig. 7). Considering the long-term climatology of double tropopauses (Fig. 6), the observed trends imply that the high-frequency double tropopause belt in the subtropics and midlatitudes of each hemisphere is expanding poleward over time. The ERA-Interim, JRA-55, and MERRA-2 reanalyses showed consistent regions of long-term increasing double tropopause frequency trends, but underestimated the poleward extent of these trends compared to the observations. As found in the primary tropopause altitude trend analyses, CFSR showed trends in double tropopause frequency that were largely inconsistent with the remaining reanalyses and observations.

Given the relationship between double tropopauses, Rossby wavebreaking, and STE above the subtropical jets, the increasing frequency of double tropopauses over time implies that similar increases in Rossby wave breaking and STE have occurred during this time period and are mostly poleward. The tropopause break-relative trend analysis (Fig. 9) further confirmed that these increasing trends are almost entirely on the poleward side of the tropopause break in each hemisphere. Consistent increases in Rossby wave breaking and transport have been found in recent studies. In particular, increases in Rossby wave breaking frequency using MERRA-2 output at an altitude between the primary and secondary tropopauses, where one would expect the closest relationship between Rossby wave breaking and the occurrence of double tropopauses, have been documented in Jing and Banerjee (2018). Modeling studies suggest that transport of air from the tropics into the extratropical lower stratosphere has also increased in both hemispheres, which has been related to recently observed decreases in lower stratospheric ozone in the extratropics (Ball et al., 2018; Wargan et al., 2018). In comparison, the spatially limited increasing trends for double tropopause frequency found equatorward of the tropopause break in this study may indicate an increase in equatorward transport of stratospheric air from the extratropics into the tropical upper troposphere (Liu and Barnes, 2018), but more work is needed to better understand the impact of these tropopause changes.

Recognizing the lack of long-term radiosonde observations over the oceans and throughout much of the southern hemisphere, it is not surprising that reanalysis tropopause altitude errors and altitude trends differ the most in these regions. In addition to impacts directly related to data assimilation, differences in long-term trends between the reanalyses are likely the result of (1) differences in vertical grids, (2) differences in the representation of physical and dynamical process that impact both short- and long-term tropopause change, and/or (3) differences in the accuracy of multiple tropopause identification. It is not clear which of these factors is the most significant contributor to the observed differences, but differences in multiple tropopauses are likely responsible for much of the disagreement within the subtropics and high-frequency double tropopause belts in each hemisphere. In particular, since failing to identify a multiple tropopause is most often the result of misidentifying the

secondary tropopause as the primary tropopause, long-term trends may be enhanced or reduced as a result of these errors (especially if they have a time dependence). As suggested by Gettelman and Wang (2015), the primary tropopause inversion layer depth has been changing over time, increasing in some regions and decreasing in others. Such changes will limit accurate identification of primary lapse-rate tropopause altitudes in the regions where it is getting shallower and improve identification

in the regions where it is getting deeper. These changes may induce false trends in primary tropopause altitude and double tropopause frequency in the reanalyses, which we have not attempted to diagnose here. Future studies should investigate the factors responsible for differences in reanalysis trends in further detail.

In summary, this work has shown that global tropopause altitudes and the frequency of double tropopauses have largely increased between 1981 and 2015. These changes are relevant to climate and UTLS composition since increases in primary

tropopause altitude are believed to be associated with a warming climate and double tropopause events often provide a physical indication of STE between the tropical upper troposphere and extratropical lower stratosphere. Broad agreement between three out of four of the modern reanalyses included in this study provides some confidence in their depictions of UTLS change. In addition, the consistency between reanalysis tropopause identifications and those from available radiosonde observations suggest that the tropopause and its behavior are well represented in modern reanalyses. Future work is needed to examine

long-term variability and trends in tropopause characteristics using additional observations and models, including existing model output from future climate projections. Longer time periods and a greater number of potential solutions from available models may provide increased confidence in the sign, magnitudes and locations of the trends diagnosed in this study and those projected to occur in the future.

*Acknowledgements.* This work is funded by the National Natural Science Foundation of China Grant 41505033 and China Scholarship

Council. The authors thank the agencies that provided reanalysis data used in this study: ERA-Interim from the European Centre for Medium Range Weather Forecasts (ECMWF) and JRA-55 from the Japan Meteorological Agency (JMA), both obtained from Reanalysis Data Archive (RDA) managed by Computational and Information Systems Laboratory (CISL) at the National Center for Atmospheric Research (NCAR), CFSR/CFSv2 from the National Centers for Environmental Prediction (NCEP), and MERRA-2 from the National Aeronautics and Space Administration (NASA) Global Modeling and Assimilation Office (GMAO). We also thank the National Oceanic and Atmospheric Ad-

ministration (NOAA) National Centers for Environmental Information (NCEI) for providing IGRA radiosonde data and the full-resolution radiosonde data used for comparison in Figure 1 and GISTEMP Team for providing GISS Surface Temperature Analysis.

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

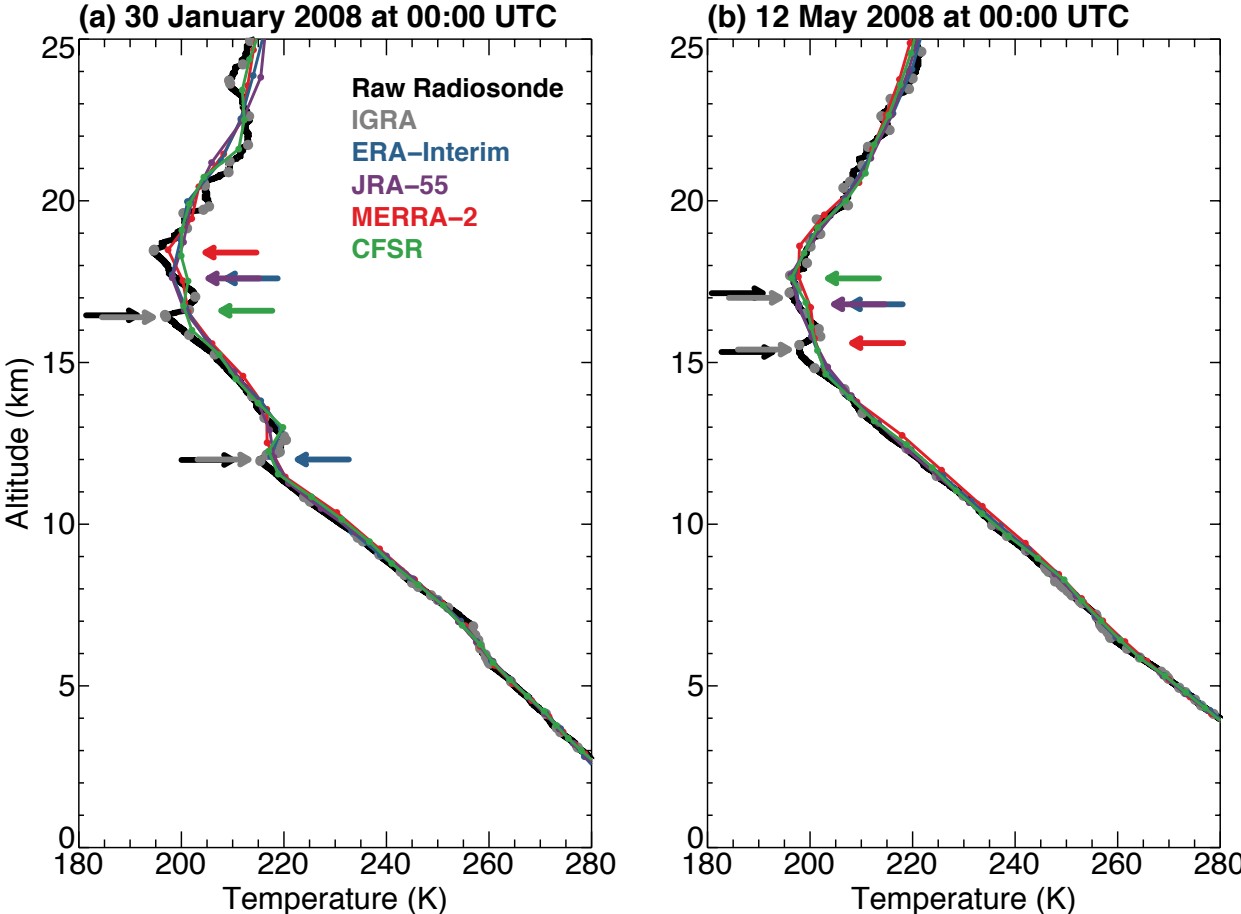

**Figure 1.** Example temperature profiles from the National Weather Service radiosonde site in Corpus Christi, Texas, at 0000 UTC on (a) 30 January 2008 and (b) 12 May 2008. Black lines show the original, full high-resolution radiosonde profile and gray dots show the reduced levels saved in the IGRA data. Coincident temperature profiles from ERA-Interim are shown in blue, JRA-55 in purple, MERRA-2 in red, and CFSR in green, with circles along these lines denoting each native model level. Colored arrows denote the locations of primary and secondary lapse-rate tropopause altitudes calculated using each temperature profile.

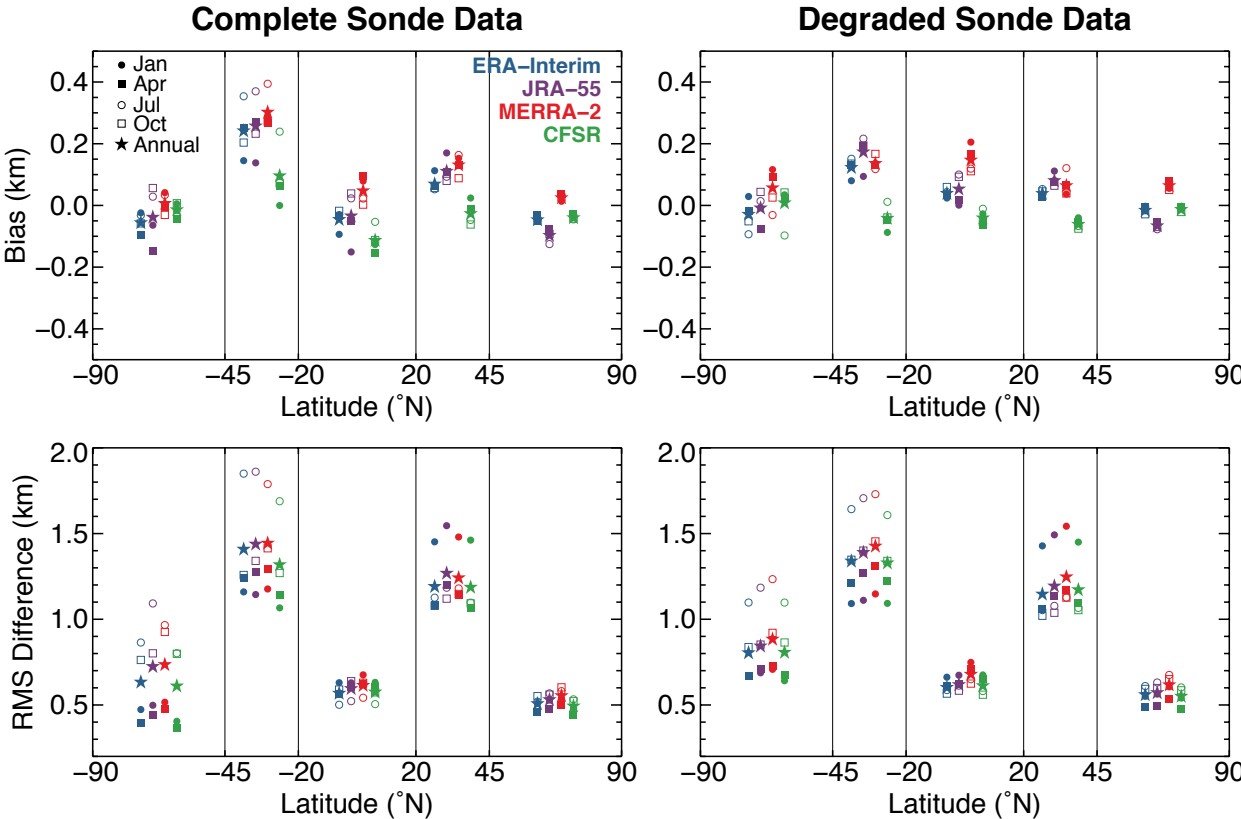

**Figure 2.** (Top panel) Average bias and (bottom panel) root-mean-square differences in instantaneous primary tropopause altitudes between radiosonde and reanalyses within five latitude bands. Open and closed symbols show the statistics as a function of season, with January (July) given as closed (open) circles, April (October) given as closed (open) squares and the annual values as closed stars. Results for ERA-Interim are shown in blue, JRA-55 in purple, MERRA-2 in red, and CFSR in green. Latitude values along the x-axis and vertical lines denote the boundaries of each latitude band. The number of radiosonde stations (profiles) used for each band are as follows: 113 ($3.5 \times 10^4$) in the northern hemisphere extratropics (45–90°N), 152 ($5.0 \times 10^4$) in the northern hemisphere subtropics (20–45°N), 25 (7,000) in the deep tropics (20°S–20°N), 20 (5,000) in the southern hemisphere subtropics (20–45°S), and 7 (2,000) in the southern hemisphere extratropics (45–90°S). Results using the complete IGRA radiosonde observations are shown on the left and results following the degradation of IGRA profiles to the native vertical grid of each reanalysis prior to tropopause calculation are shown on the right.

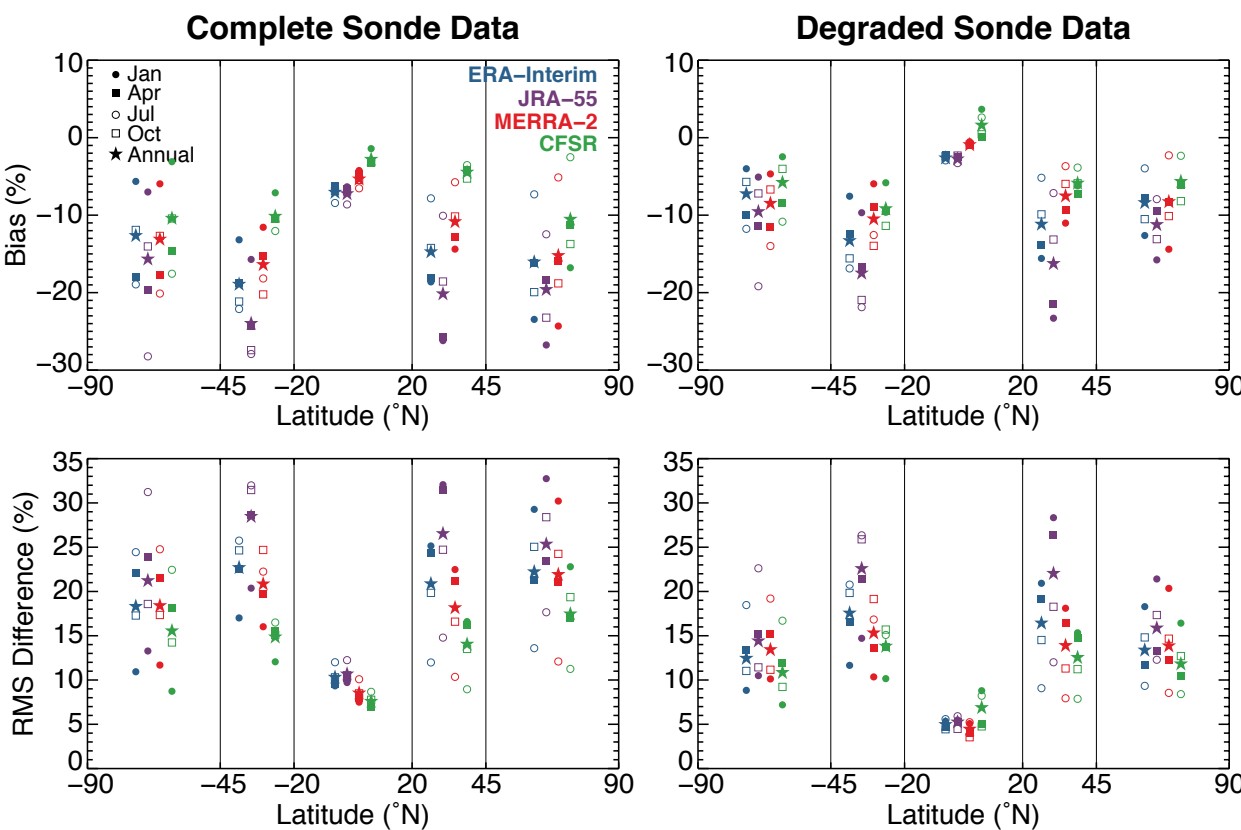

**Figure 3.** As in Fig. 2, but for the monthly double tropopause frequency.

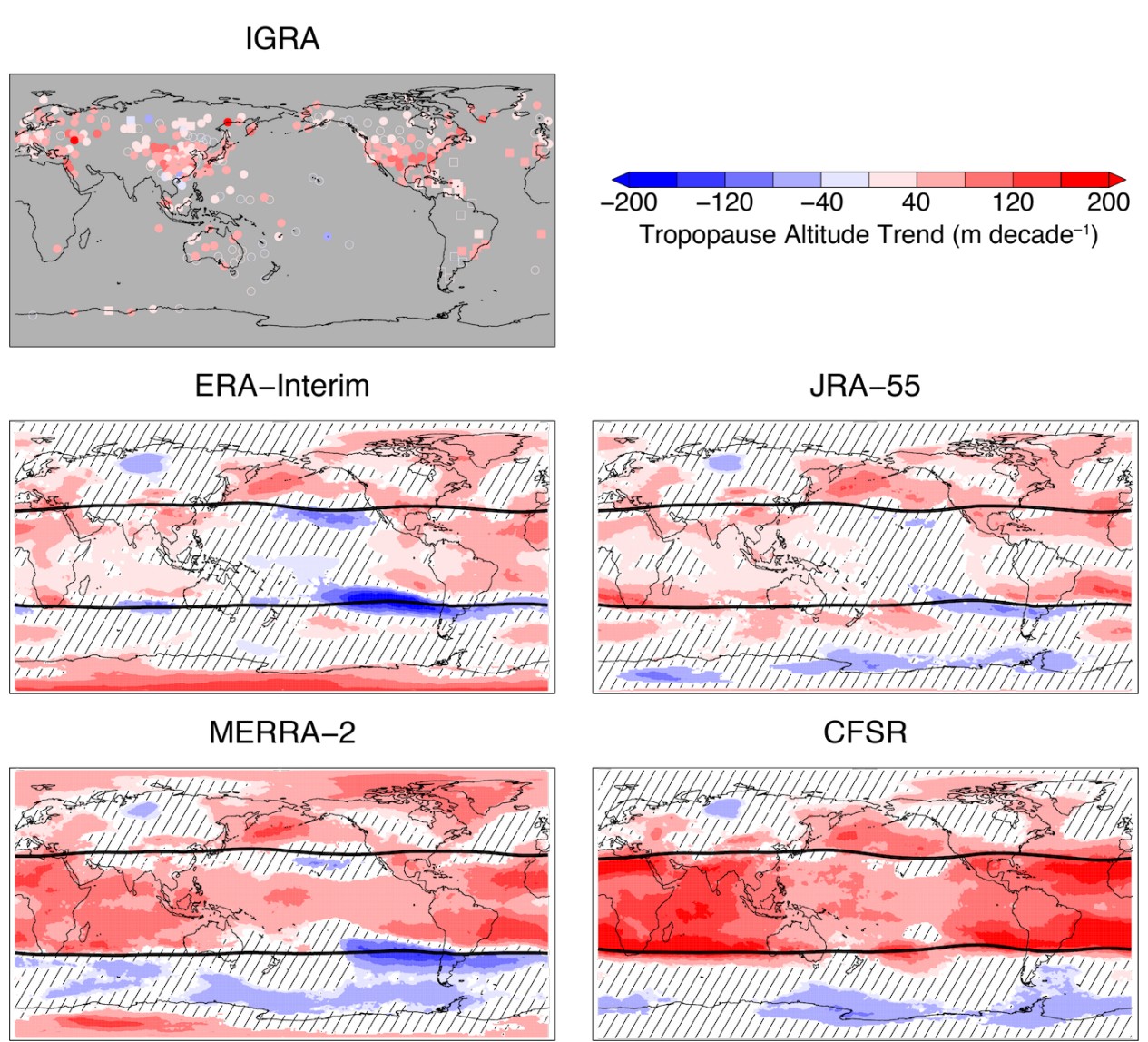

**Figure 4.** Eulerian mean trends of the primary tropopause altitude from 1981 to 2015 for IGRA radiosonde observations and the reanalyses. For IGRA trends, circles denote 0000 UTC trends and squares denote 1200 UTC trends, with filled symbols denoting statistical significance. Colored areas of the reanalysis maps are statistically significant, while line-filled regions are not. Thick black lines in each reanalysis map show the 35-year mean tropopause break latitudes.

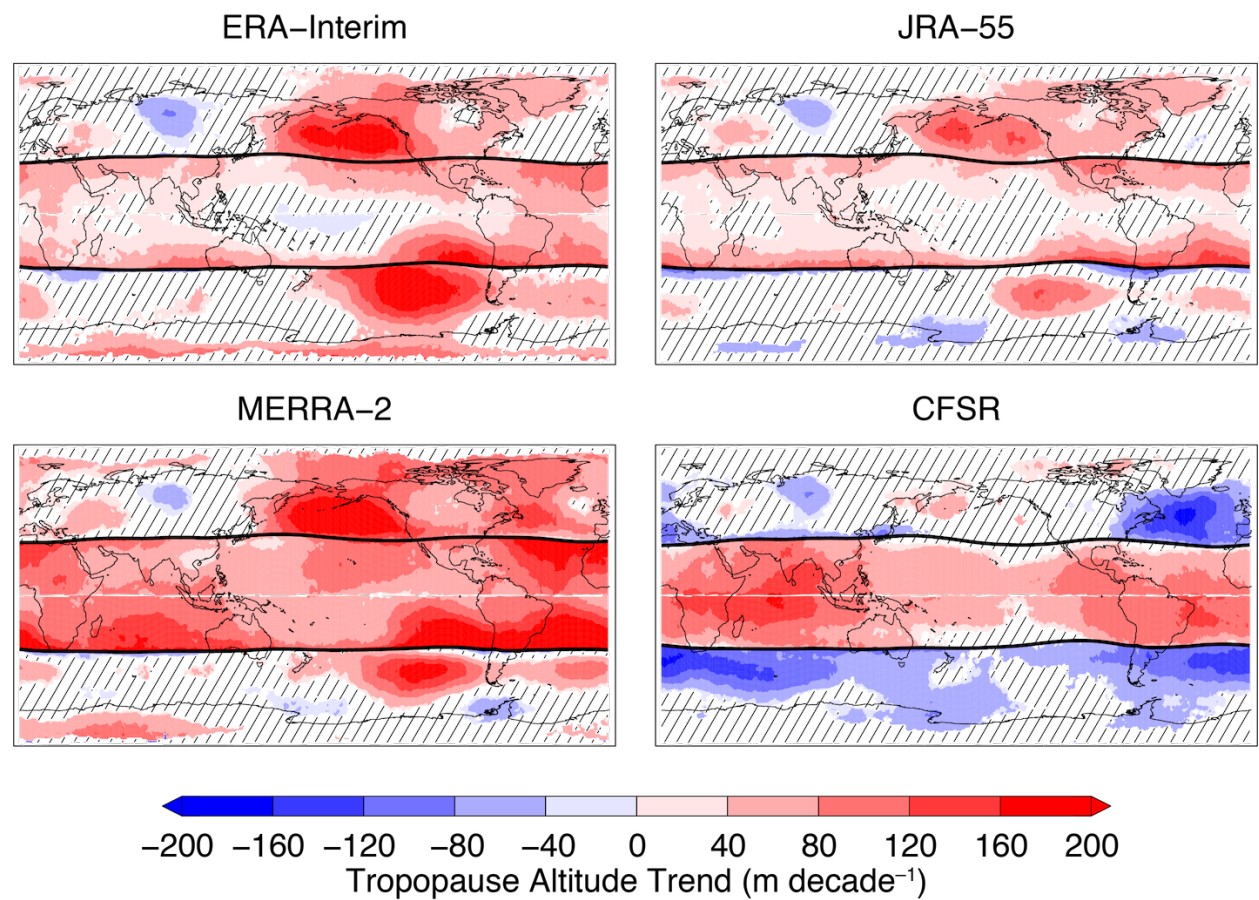

**Figure 5.** As in Fig. 4, but for tropopause-break relative primary tropopause altitudes from reanalyses only.

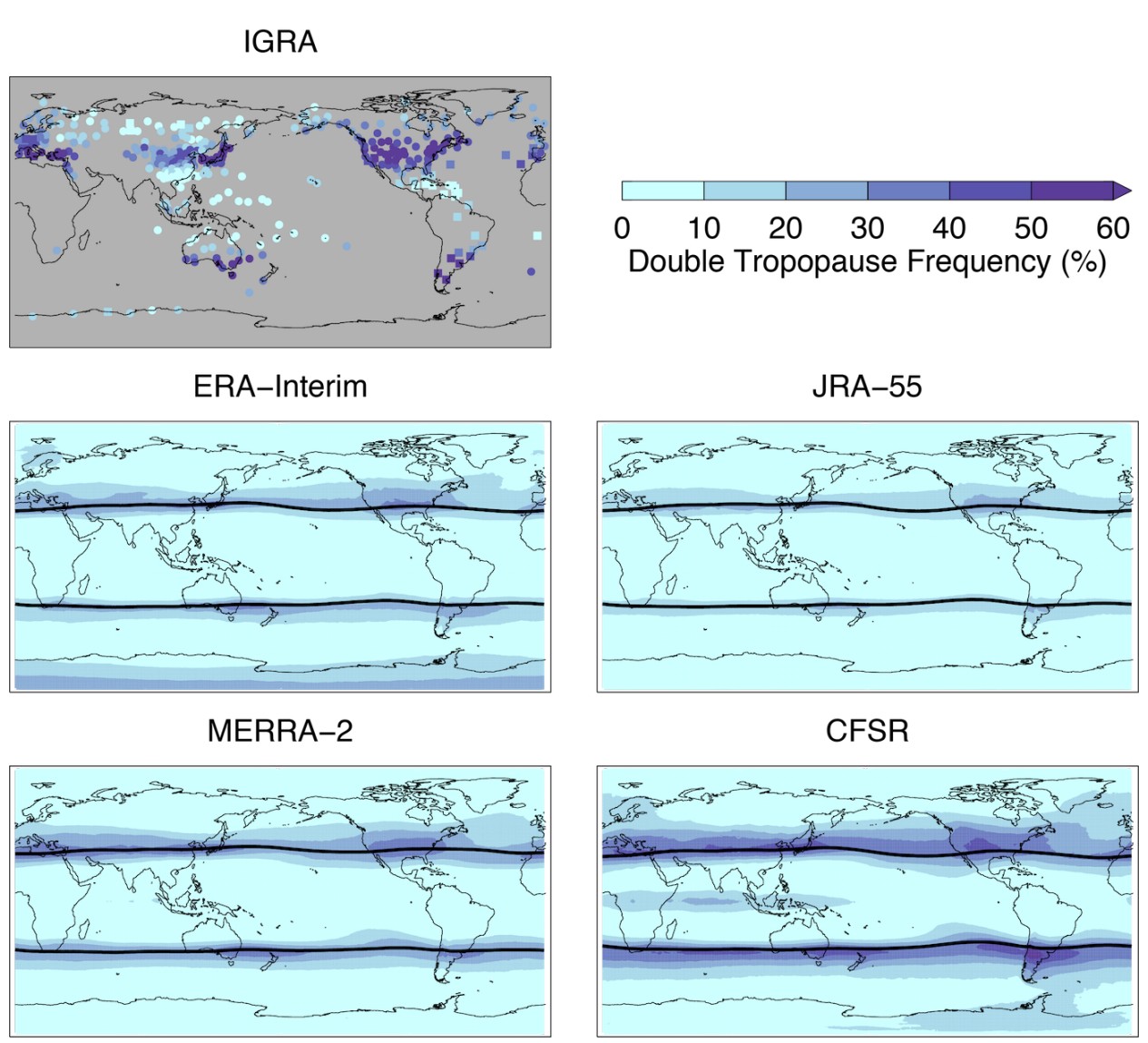

**Figure 6.** Average double tropopause frequency from 1981 to 2015 for IGRA radiosonde observations and the reanalyses. For IGRA, circles denote 0000 UTC frequencies and squares denote 1200 UTC frequencies. Thick black lines in each reanalysis map show the 35-year mean tropopause break latitudes.

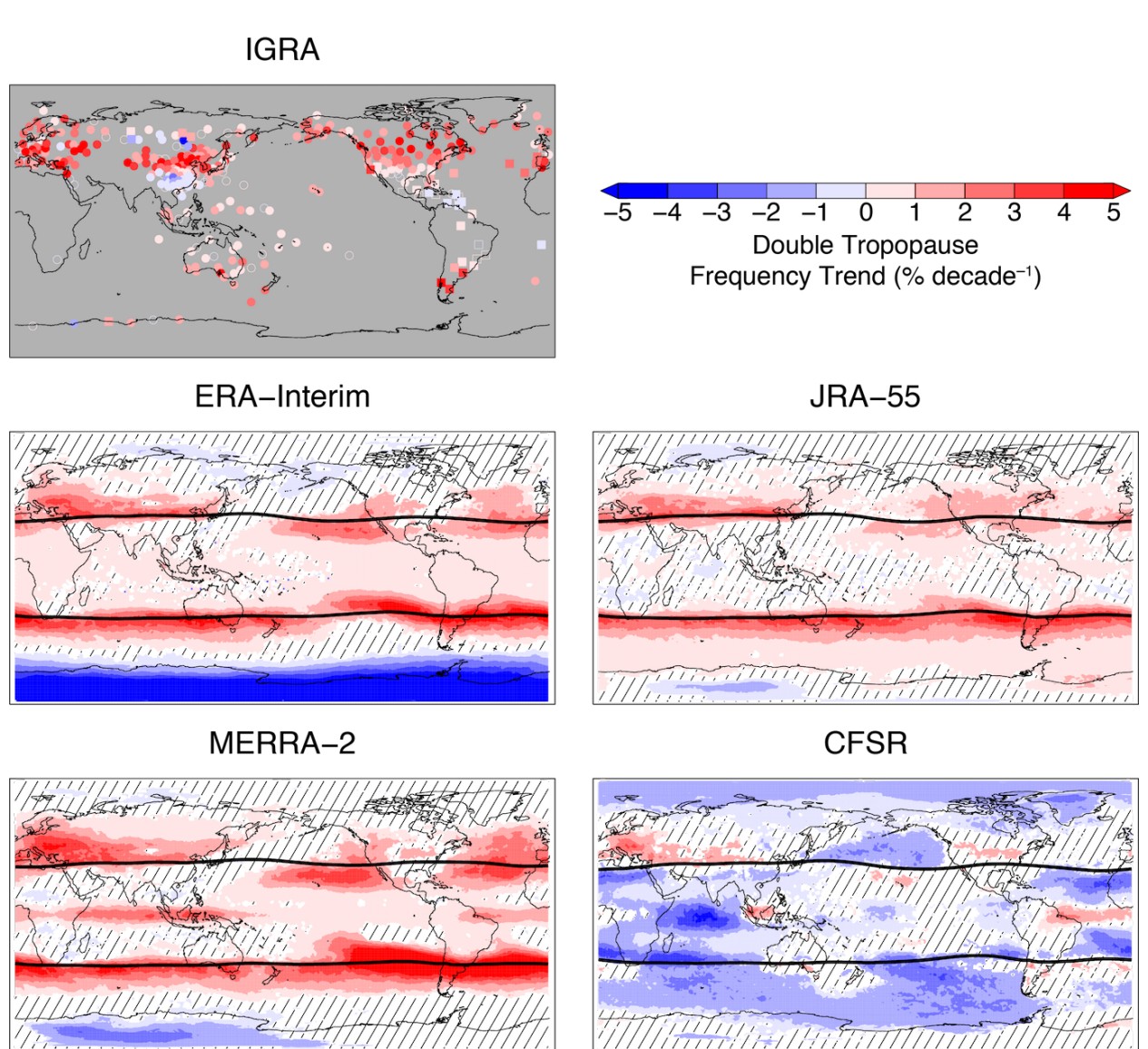

**Figure 7.** As in Fig. 4, but for trends in double tropopause frequency.

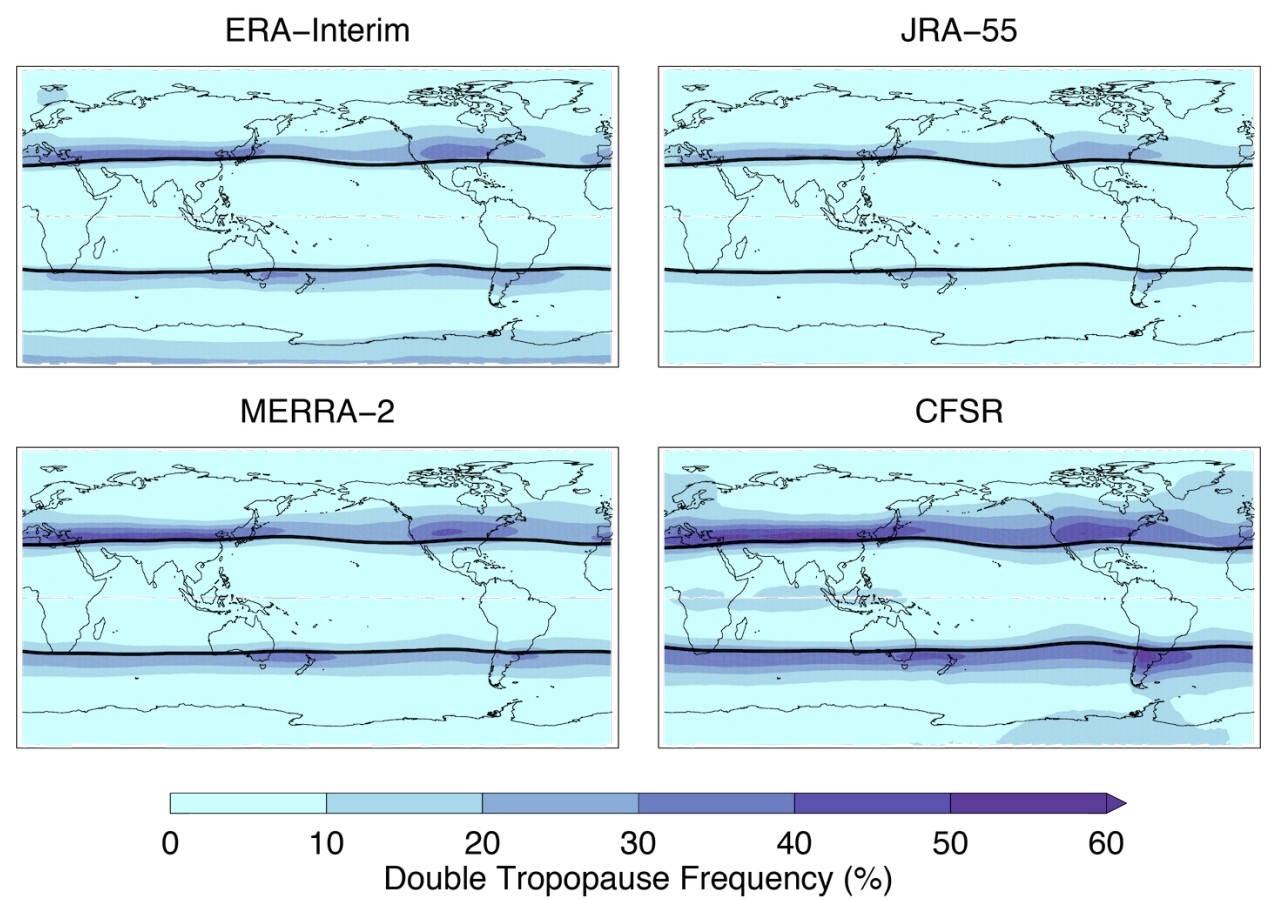

**Figure 8.** As in Fig. 6, but for reanalyses only in a tropopause break-relative coordinate.

## ERA–Interim

## JRA–55

## MERRA–2

## CFSR

Double Tropopause Frequency Trend (% decade⁻¹)

**Figure 9.** As in Fig. 7, but for reanalyses only in a tropopause break-relative coordinate.

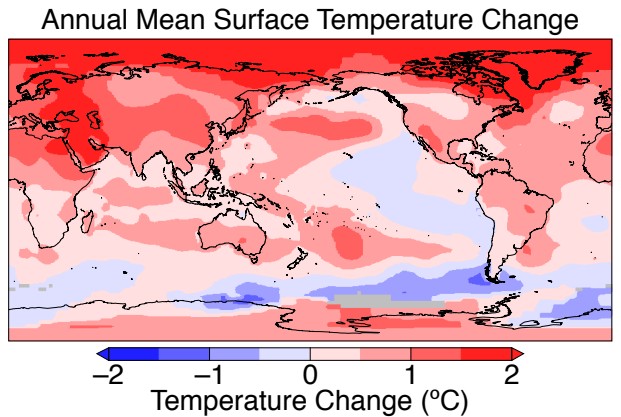

## Annual Mean Surface Temperature Change

Temperature Change (ºC)

**Figure 10.** Observed 35-year (1981-2015) annual mean surface temperature changes from the NASA GISS surface temperature analysis.

**Table 1.** Bias and root-mean-square differences between reanalysis tropopause identifications and radiosonde tropopause identifications. Bias is defined as reanalysis − radiosonde, with the frequencies of positive and negative bias in parentheses. The number of radiosonde profiles used for primary tropopause altitudes and double tropopause frequencies is 99,023 and 45,181, respectively.

| | Primary Tropopause Altitude | | | Double Tropopause Frequency | | |
|---|---|---|---|---|---|---|
| | Positive Bias (km) | Negative Bias (km) | RMS difference (km) | Positive Bias (%) | Negative Bias (%) | RMS difference (%) |
| **Complete Sonde Data** | | | | | | |
| ERA-Interim | 0.584 (45.589 %) | -0.436 (54.411 %) | 0.973 | 2.156 (18.816 %) | -18.696 (81.184 %) | 20.922 |
| JRA-55 | 0.657 (43.565 %) | -0.447 (56.435 %) | 1.030 | 0.684 (14.786 %) | -22.414 (85.214 %) | 25.291 |
| MERRA-2 | 0.599 (54.552 %) | -0.506 (45.448 %) | 1.019 | 3.725 (23.332 %) | -17.271 (76.668 %) | 19.386 |
| CFSR | 0.508 (45.944 %) | -0.485 (54.057 %) | 0.961 | 6.525 (32.662 %) | -13.570 (67.336 %) | 15.200 |
| **Degraded Sonde Data** | | | | | | |
| ERA-Interim | 0.464 (50.844 %) | -0.428 (49.156 %) | 0.939 | 1.117 (32.772 %) | -14.454 (67.228 %) | 14.720 |
| JRA-55 | 0.538 (47.669 %) | -0.429 (52.331 %) | 0.974 | 0.265 (26.641 %) | -17.877 (73.359 %) | 18.912 |
| MERRA-2 | 0.515 (57.610 %) | -0.511 (42.391 %) | 1.021 | 2.109 (39.919 %) | -13.561 (60.081 %) | 13.451 |
| CFSR | 0.437 (49.782 %) | -0.507 (50.218 %) | 0.950 | 3.421 (44.586 %) | -12.193 (55.414 %) | 12.001 |

The values given are root-mean-square differences and mean differences between reanalyses and radiosonde observations.