# Peer review of "Global Tropopause Altitudes in Radiosondes and Reanalyses"

_Atmospheric Chemistry and Physics, 2018_

## Short Comment (SC1) · 8 Nov 2018

Although I have not been formally invited to review this paper, as it is line with a big part of my research along the last decade I have read it carefully and I have some comments that I would like to share with the authors. They could want to include them in the final version.

First of all I would like to congratulate Xian and Homeyer for this valuable effort. This analysis brings some light on reanalysis bias at UTLS levels and updates previous results existing in the literature.

Also I would like to clarify that although I mention in my comments several of my previous works, my intention is to highlight some points that can help to support part of the

results here exposed and to bring a balance to the discussion. I miss in the text some punctual comparisons with previous works (many of them overlooked in the current version and in some way creating a misrepresentation of the topic) that I think would help us as readers to get a better picture of the research topic, instead of having to go through different papers.

More specifically:

- in page 1 line 23 I miss a citation to Añel et al. (2006). This work also deals with the trends from radiosonde data and indeed it will be useful to discuss some issues later in the paper;

============================================================

- in page 2, after line 17: usually there is some confusion on the issue of definition of the tropopause. Words have meanings and being fair it only exists one definition for the tropopause, the one established in 1957 by the WMO. Others are 'criteria' to approach the behavior of the 'tropopause' or UTLS transition according to the best fit for different studies, campaigns, etc. This does not change the reality of the complex atmospheric behavior, but using the right words is useful for those not so familiar with the topic that could waste time looking for formal definitions that do not exist anywhere. Therefore in line 18 it is not 'The conventional tropopause definition' but 'The tropopause definition';

============================================================

- in page 2, line 22: in some way linked with the previous issue, I do not think that it is correct to say that there are exceptions to 'performance'. Simply there are regions of the Earth where the UTLS structure is so complex that there is not a tropopause or transition troposphere/stratosphere as such. You mention one case where this behavior is mostly driven by the very specific tropospheric radiative balance during the austral winter. But it is not the only case. The same happens in the third-pole (the Tibetan Plateau) but because of dynamical reasons. There unstable mix of air can make

impossible to get a troposphere-stratosphere distinction because of the high altitude of the plateau and its radiative balance (see Chen et al. 2013 and Chen et al. 2016);

=============================================================

- page 3 line 13: indeed fifteen years before Hoinka et al. (1998) had clearly established that the usual values of 1.6 PVU introduced in a campaing in the 1980's or the 'popular' 2 PVU value underestimate the reality of the tropopause height (obviously in extratropics and polar regions);

=============================================================

- subsection 2.1 "Reanalysis output": for the purpose of this work, more relevant than this information (vertical levels and top) is to know the distribution of levels (or dz) between 200 hPa and 50 hPa. I would recommend to the authors to focus the description here on this layer. This will enable them to simplify the understanding and discussion of results later, for example in section 3;

=============================================================

- page 6 lines 5-10: this is a good exercise to guarantee representation with a case study. But this had already been proved by Antuña et al. (2006) using other station at a quite similar geographical location. I recommend to cite the work to add extra support and to include in the text the coordinates for Corpus Christi (unless I have missed them);

=============================================================

- in subsection 2.4 you state 'the 35-year analysis period'. I have not got clearly what is the period of study: 1979-2015? This is 37 years. 1981-2015?. Please, clarify it;

=============================================================

- page 8, lines 15-16: there is another basis for this (one of them briefly mentioned in the paper), the competing phenomena of tropical widening where the tropical
tropopause overlaps the extratropical one and the horizontal meridional entraintment of extratropical air to tropical regions (Wang and Polvani, 2011; Añel et al. 2012; Castanheira and Gimeno, 2011).

==============================================================

- subsection 3.2, first paragraph: this is in agreement with the results for the Scenario 1 studied by Añel et al. (2006). That is, raw series without data homogenization. Thought IGRA solved several of the problems that existed in CARDS, here you do not perform any change-point detection technique and this restricts the validity of your results. I think that the issue of not undergoing change-point detection deserves to be mentioned here and that a comparison in the text with the values obtained by Añel et al. (2006) and Santer et al. (2003a,b) would be good as it would enable readers to get a more complete picture of the state-of-the-art.

The point on Siberia deserves special attention in my view: this is also in agreement for with part of the Scenario 1, and with Scenarios 2 and 3 of Añel et al. (2006). Here I would point out two different issues:

1. some of the radiosonde series in this region show up to a 1% significant correlation with the Northern Annular Mode, this could explain partial regional trends. But as soon as in the 1960's Makhover reported that this region has a special behavior in comparison with similar latitudes in this hemisphere (check Antuña et al. 2009 or the original Russian books cited therein);

2. no doubt it deserves a deeper analysis with data homogenization techniques, but there is a potential reason that could explain bias (be aware that I talk about bias not changes in trends) over the region corresponding to the former Soviet Union. This reason is the use of different radiosondes with very different equipment than the extended Vaisala RS80/RS90 radiosondes for other parts of the world. A quick check of the metadata in IGRA shows how some stations over the period 1980-1990 there was up to 4 or 5 changes of radiosonde model (changes, not simple updates) and in some of

them radiation corrections in 90's. This kind of problems with soundings over Russian territory with frequent radiation corrections was also pointed out by Makhover (again see Antuña et al. 2009). This could have an impact on any trend computed. Therefore any statement on trends without change point detection and data homogenization should be accompanied of one on the limitations of the data analysis.

==============================================================

- subsection 3.3, last sentence: I think that it could exist a partial explanation for this behavior in Fig.4 for CFSR. This is my hypothesis: as it has been proved by Añel et al. (2008) in presence of multiple tropopauses the first lapse rate tropopause (LRT1) is lower than when a single tropopause exist and multiple tropopauses are not present. As Xian and Homeyer show CFSR has lower bias and increased resolution at UTLS levels. This enables this dataset to better represent a bigger number of multiple tropopause events. Having more multiple tropopause events means that an increasing proportion of lower LRT1 cases should be found. This should be more clear in critical regions for the detection, such as subtropics. Therefore the positive trend in the frequency of multiple tropopauses and lower bias of CFSR would be driven an increased frequency of lower LRT1.

==============================================================

- page 10, lines 9-15: this is exactly what is stated in Castanheira et al. (2009) (Fig. 8) using IGRA data and a probable consequence of the energetic modes at UTLS levels. I think that the numbers here obtained should be compared to their ones and the work cited.

==============================================================

- page 13, line 14: I do not think that "found" is the right word here. To be fair beyond the useful contribution on comparison between state-of-the-art reanalysis, the other results here presented only confirm previous findings existing in the literature and it

should be acknowledge in this way.

===============================================================

- Table 1: I understand that values in this table are computed using all the stations, independently of the hemisphere. This could provide a sense of average changes, but if you present the results for months representative of seasons, what is the point on mixing NH and SH stations?. Doing such thing does not let to appreciate the true seasonal change. In my view exposing only the values for extratropical regions of one of the hemispheres would be the right way of doing it, as there is no point on including the tropics because of the lack of seasonal variability. Moreover Double tropopauses are a phenomenon with strong seasonal dependence associated to extratropical wintertime UTLS baroclinicity (Castanheira et al. 2009) and therefore the same reasoning applies.

===============================================================

**References**:

Antuña et al. (2006) Impact of missing sounding reports on mandatory levels and tropopause statistics: a case study, Ann. Geophys., 24, 2445–2449.

Antuña et al. (2009) Professor Zalman Makhover: a relevant contributor to early tropopause studies, Meteorol. Zeit., 18(6) 573-584.

Añel et al. (2006) Changes in tropopause height for the Eurasian region determined from CARDS radiosonde data, Naturw. 93:603–609, DOI 10.1007/s00114-006-0147-5

Añel et al. (2012) On the Origin of the Air between Multiple Tropopauses at Midlatitudes, The Scientific World Journal, vol. 2012, Article ID 191028, 5 pages. DOI: 10.1100/2012/191028.

Castanheira et al. (2009) Increase of upper troposphere/lower stratosphere wave baroclinicity during the second half of the 20th century, Atmos. Chem. Phys., 9, 9143-9153, DOI: 10.5194/acp-9-9143-2009.

Castanheira, J. M., and L. Gimeno (2011), Association of double tropopause events with baroclinic waves, J. Geophys. Res., 116, D19113, doi: 10.1029/2011JD016163.

Chen et al. (2013) The Deep Atmospheric Boundary Layer and Its Significance to the Stratosphere and Troposphere Exchange over the Tibetan Plateau, PLoS ONE, 8(2): e56909. doi:10.1371/journal.pone.0056909

Chen et al. (2016) Reasons for the Extremely High-Ranging Planetary Boundary Layer over the Western Tibetan Plateau in Winter, J. Atmos. Sci., 73, 2021-2038. DOI: 10.1175/JAS-D-15-0148.1

Hoinka et al. (1998) Statistics of the Global Tropopause Pressure, Mon. Wea. Rev., 126, 3303–3325.

Ribera et al. (1998) Quasi-biennial modulation of the Northern Hemisphere tropopause height and temperature, J. Geophys. Res., 113, D00B02, doi: 10.1029/2007JD009765.

Wang, S., and L. M. Polvani (2011), Double tropopause formation in idealized baroclinic life cycles: The key role of an initial tropopause inversion layer, J. Geophys. Res., 116, D05108, doi: 10.1029/2010JD015118.

---

## Referee Comment (RC1) · Anonymous Referee #1 · 17 Nov 2018

This paper documents the lapse-rate primary and double tropopause reported in the most current reanalyses. The purpose of the paper is suitable for ACP, especially for the SRIP special issue. However, I request a minor revision to the manuscript because there are a few places that need to be clarified.

In my understanding, this paper targets two parts of validation: 1) comparing to radiosonde data, how high the primary tropopause and how often the double tropopause showing in reanalysis, and 2) how are the long-term variability of tropopause inferred from the reanalyses comparing to that from radiosondes.

The first part of validation is very useful for the UTLS community because lots of analysis relies on using the tropopause level as a reference. Even though the detailed tropopause heights might be tremendously different from what's showing in the radiosondes (e.g., Fig. 1), the variability of tropopause in reanalyses might not be that different, because the variability is more or less determined by the model settings and core dynamics that are consistent along the run. This is clearly shown in the analysis that in terms of primary tropopause altitude, MERRA-2 behaves much worse than the others, but in terms of trends, MERRA-2 is not that bad at all. Therefore, the long-term trend of tropopause is potentially useful to the community connecting the UTLS studies to the big scenario of climate change – although at this point the more urgent request would be to disentangle what caused the discrepancies in trend analysis among different realayses.

I am glad that the authors took my suggestion and added Fig. 1 to the revised manuscript. This figure illustrates how the relatively coarse vertical resolution of reanalyses could have distorted the tropopause analysis. This enhances our voice for the modeling community that adding more levels around the sharp gradient (both physically and chemically) tropopause is absolutely necessary.

My major comment is that the authors should have included detailed analysis for both bias and RMS while validating. While the authors tend to emphasize the RMS statistics, I hope they know that the RMS, as a loss function, gives a relatively high weight on large errors since it is more sensitive to extreme values on long tails/outliers due to the fact that the errors are squared before they are averaged. So, sometimes very few extreme values can completely change the statistics, which is not desirable for this analysis.

In contrast, the bias is sometimes more intuitive because it tells us how much of absolute differences between the radiosonde and the reanalysis. However, the bias analysis is not perfect because the positive and negative biases will cancel out. I would separate the bias analysis into positive and negative, with positive means the reanalysis primary tropopause showing at a higher altitude, and negative means the reanalysis primary tropopause showing at a lower altitude. Meanwhile, adding the frequency (with respect to total samples considered) of positive and negative bias. In this way, we know that on average how frequent and how much the reanalysis would overestimate/underestimate

the tropopause height. I think this detailed analysis is more meaningful to the community.

In this sense, it is fair to always include both bias and RMS error analysis. I think Fig. 2 should include bias on the first panel and RMS error on the second panel. For each panel, different shapes represent different months, but please do include one more statistics for all-season averages. Then, add another figure that repeats a similar analysis for the double tropopause. Having the easy visualization of the statistics, still, keep Table 1 of detailed numbers for easy reference.

Another major comment is on the accuracy of IGRA data, and its ability to precisely document the lapse-rate tropopause is crucial for this study. It helps if the authors could iterate in more details on how the <= 50m vertical resolution of raw observation are eventually reported in only 1.5-2.5 km vertical resolution at the UTLS (although in the revised manuscript the authors changed to > 1km). The "> 1 km" is still less desirable for studying the vertical variability of temperature records - it will miss effects of both gravity waves and the Rossby waves acting on the temperatures. Given the reported resolution, why not using GPS/COSMIC temperature records that has a better coverage? The focus of the paper is from 5-20 km, in which COSMIC is totally capable of seeing waves on temperatures.

If I understand it correctly, Figs. 1-2 and Table 1 are the only places that the authors performed apple-to-apple comparison by collocating the reanalysis to the radiosonde locations. For all other analysis, the authors just reported trends inferred from gridded results at each latitude-longitude box, so the results could be biased by sampling sizes. So, the first part of validation is more meaningful to my sense. That said, personally I am not interested in the trends reported. For example, what does a trend of +/- 50 m/decade in primary tropopause mean? What does a positive trend of double tropopause frequency mean in specific reanalysis? Unless you can elucidate the possible cause of trends with proof, I don't think the trend numbers themselves have significant meaning. On the positive side, the fact that different reanalyses showing

different trends is meaningful in that they imply how unreliable the reanalyses are as to the tropopause analysis. This makes me wondering if it is necessary to include the trend analysis, especially in such a large portion of the paper. If I were the authors I would report the bias and errors in more details to help the community to understand the different performances of the reanalyses.

A last comment is that I do hope the authors could put more emphasis on the physical meaning/causes of the (large) differences among different reanalysis. So, beyond the vertical resolution, could there be any other reasons that caused the discrepancies? The current version seems to be less scientific and more like a technical report.

Minor wording comments: 1. P1L8, attributed –> attributable 2. P1L9: observations 3. P1L9: and reanalyses –> and the reanalyses 4. P1L9: analysis period 5. P2L10: –> the UTLS composition 6. P1L13-15: this sentence doesn't make sense 7. P3L6-7: this makes sense because of the existence of the ozone layer, but can you be more specific about it? 8. P4L1: I don't understand the logic here. If the radiosonde data is so limited, why bothering using them instead of COSMIC data? Plus, this part sounds like belonging to the discussion part. 9. P5L10-12: all reanalyses are reported in sigma or eta coordinates. From conversion you might get temperatures on pressure levels easily, but how did you get them in altitude coordinates? Did you use simultaneous geopotential heights to interpolate the data? Be more specific about how you preprocessed the data. 10. P5L16: –> quality-controlled 11. P5L23-24: one comment is that this linear interpolation doesn't change the shape of the profile, at all. So, you typically end up with the same value without doing interpolation. 12. P7L18: tropopause altitudes in MERRA-2 → primary tropopause altitudes in MERRA-2. 13. P8L16: is maximum –> its maximum 14. P13L11-12: how did you reach this conclusion?

---

## Short Comment (SC2) · 14 Dec 2018

I found this study quite interesting for the ongoing research on the tropopause region in a climate perspective. It gives considerable consistency to previous evidences indicating a positive decadal trend both in the global tropopause height and in the extratropical double-tropopause frequency – at the expense of stressing the discrepancies between the results inferred from reanalysis products and RAOB, as well as showing differences among different reanalysis models.

I also think that the paper could be improved and hope that the comments below can be valuable in the eyes of the authors.

Main aspects:

[Figure]

1) Concerning the reproducibility of results, the paper lacks information about the radiosonde data used in the study: how were IGRA stations selected in the first place? The number of selected stations (317) and the corresponding amount of observations for 1985-2015 are given later in the results section, with their approximate locations shown in the Figures. But IGRA (version 2 released in 2016) contains temperature data from 800–900 radiosonde stations within the studied period. Nothing, however, is said about the choice of stations, concerning the homogeneity of time-series in terms of temporal and vertical features (i.e., leaving aside the much more difficult problem of instrument biases): temporal regularity and continuity; vertical resolution around the tropopause.

2) A linear interpolation to a 200-m regular vertical grid was applied prior both to radiosonde and reanalysis temperature data before tropopause identification. The authors claim this was done "in order to enable reliable tropopause identification". This phrase is potentially confusing to the reader. Evidently, an interpolation is needed to verify the second condition of WMO's definition of first tropopause, as well as to look for a second tropopause. But a linear interpolation simply does not change the lapse rate between the known data points. So, the estimation of the first and second tropopause levels is essentially limited by the resolution of data – as the authors in fact recognize in other parts of the paper. The gain resulting from the interpolation scheme should be explained to make this point clear.

3) Radiosonde data were analyzed at the principal synoptic hours, 0000UT and 1200UT, whereas reanalysis data were analyzed only at 0000UT. This means that half of the time-zones on the global reanalysis fields of temperature (at latitudes outside of the polar regions, after averaging over one or more years) is represented by daylight times, while the other half is represented by nocturnal times. In this respect, in Figs. 3-6 it is not clear why some radiosonde stations show 0000UT average values while others show 0012UT values, since reanalysis-derived values refer always to 0000UT. Also, considering the diurnal variations of the tropopause height, it should be explained

how the radiosonde–reanalysis tropopause differences listed in Table 1 were exactly calculated.

4) Although not obligatory, to be more informative Table 1 should depict hemispheric seasons. Or perhaps individual months, but then restricting to North Hemisphere, where the amount of radiosonde data (used as reference to errors) is much larger there than in the South Hemisphere.

5) The calculation of tropopause altitude needs a bit of clarification: is moisture included in the hypsometric equation? 'Tropopause altitude' refers to geometric altitude or geopotential altitude?

6) Maybe the large discrepancies between the results obtained from CFSR and the other reanalysis models (seen in all plots) deserve a slight explanation.

Secondary aspects:

P2, L19. Where it reads "(. . .) (also known as vertical temperature gradient) (. . .) " it correctly should read "(. . .) (negative of the vertical temperature gradient) (. . .)"

P4, L1. "(. . .) since they are only launched from land masses". Considering the radiosondes launched on whether ships and 'ships of opportunity' (even if not used in the study) it should be better to write "(. . .) since they are mostly launched from land-masses".

P4, L6. "Reanalyses assimilate global high-quality observations (. . .)". Do not forget to mention other observation platforms besides radiosondes. Moreover, I doubt that all observations assimilated in reanalysis models are of "high-quality". A meteorological reanalysis is supposed to deal with inaccurate and incomplete observations to some degree. "Quality-controlled" is closer to reality.

P4, L 30. I don't understand the words "a physical perspective of the UTLS'. I suppose that the authors' point is that their paper provides an evaluation of reanalysis-model performance regarding the UTLS temperature structure.

P6, L12. "Thus, we are confident that IGRA data are suitable for tropopause analyses following the methods employed here." How can you tell, from a demonstration with two random soundings from one site? The study uses nearly 10ˆ5 soundings from over 300 radiosonde stations! The above assertion is not acceptable. Although Fig. 1 serves the purpose of illustration of the idea, paradoxically, expressing here some uncertainty would give more confidence to the reader.

P7, L14-16. It's not totally clear whether Fig. 3 (and so on) uses only four months per year or not.

P13, L16. "(...) increases in primary tropopause altitude are associated with a warming climate (...)". The suggested connection is supported by a very few modeling experiments until now. I'd replace "are" by something less assertive like "is probably" or "is believed to be".

Fig. 6 and Fig. 8. If possible, the color scale legend "Double tropopause frequency" should be changed to "Double tropopause trend".

———————————————

---

## Referee Comment (RC2) · Anonymous Referee #2 · 27 Dec 2018

This paper presents a comparison of tropopause heights and double tropopause frequency between radiosondes and modern reanalyses. Most modern reanalyses on average reproduce the radiosonde tropopause heights very well, except for MERRA-2, which is biased high by $\sim$0.5 km. Double tropopause frequency is generally underestimated by the reanalyses, as expected due to their relatively coarse vertical resolution. In overall agreement with previous estimates, an upward tropopause trend is found based on radiosondes and reanalyses. Double tropopause frequency is found to show an increasing climate trend. The authors also show results based on a tropopause-break relative latitude coordinate, which shows sharper structures in the subtropics and is useful for deducing tropopause-based tropical width trends.

Overall this is a nice study, which can serve as important reference for future estimates

of tropopause characteristics and associated trends. My main suggestion is to try harder to interpret some of the findings, see detailed comment below. In its current form the paper is very descriptive and leaves almost all of the deeper understanding of its main results to future studies, which is fine for some results, but others would strongly benefit from a bit more analysis, which I'd think is meaningful within the scopes of the current paper.

I have two general comments that may require extra work, which is why I've recommended "major revisions". See further below for a list of minor comments, including of editorial nature.

General comments:

1) The fairly large bias in MERRA-2 is interesting and I was surprised that it didn't receive more attention by the authors (at least not in the writeup). After all, this is a (re)analysis, i.e., it includes a modern data assimilation scheme, presumably assimilating the radiosonde observations that here used as a reference. So my expectation was that all modern reanalyses essentially reproduce the tropopause. Fig. 1 furthermore stimulates suspicion: how can a reanalysis have such large temperature biases (> 5 K!!) in the upper troposphere? Without labelling I would have guessed that this is a free-running model. Don't you expect all modern reanalyses to very closely agree about temperature in the upper troposphere? This is the case between the other three products: ERA-Interim, JRA-55, CFSR. Is this simply an outlier example or do you often find such large biases in MERRA-2? Is this something that's documented in the literature? To be honest, if this is a robust bias in MERRA-2, then this product shouldn't be used for UTLS studies . . . in any case, this requires more discussion by the authors.

2) Vertical resolution is mentioned at many places to potentially explain differences between radiosondes and reanalyses. Isn't this easily testable? You could degrade the radiosondes to the model resolutions and see if that really explains the differences. You could even study some of the characteristics (e.g., double tropopause frequency) as

a function of vertical resolution by gradually degrading the radiosonde data. Perhaps the authors have already tried this, in any case, I would strongly suggest to include corresponding results / discussion in the paper.

Minor comments:

page 2, line 16: "uncertainty that is comparable to the vertical resolution of the model" — this makes intuitive sense, but is this a priori clear given that you interpolate between levels for the tropopause calculation?

page 2, line 19: the lapse rate is equal to _minus_ the vertical temperature gradient

page 2, line 28-29: Anel ref's

page 3, line 7-8: sentence doesn't work like this; how about: "PV, which is conserved . . ., is commonly used for transport studies in the extratropics and often used to define a dynamical tropopause . . ."

page 3, line 10: "threshold used varies considerably" — seems like an exaggeration (I'd suggest to remove "considerably"), note a lot of the STE studies (e.g., Wernli group and others) use 2 PVU and this value seems to be used mostly

page 5, lines 11-12: these are somewhat subjective choices — have you checked the corresponding sensitivity? E.g., are the results sensitive to obtaining tropopause levels from the native horizontal and vertical grid, and interpolating to the 1-by-1 lon-lat grid afterwards? I'm also not sure I understand the purpose of oversampling to the 200-m grid in the vertical for tropopause identification — please provide rationale (relevant for line 24 as well).

page 6, line 24: how do you assess whether data points are roughly evenly distributed?

page 7, line 8-9: do you do this separately for the two hemispheres? How do you then handle the equator, which in the relative coordinates "moves" around?

page 7, line 13-14: so here you suggest that you do use the native model grids for

tropopause calculations, in contrast to the description on page 5 – please clarify

page 7, line 18: is this bias a function of latitude?

page 8, discussion of Fig. 2: have you considered normalizing these RMS differences by a measure of internal/natural variability (e.g., interannual standard deviation)? Larger RMS differences would be expected in regions with larger internal variability, so part of the latitudinal differences could be related to different internal variability.

page 9, line 5: over the Atlantic trends are larger at the edges of the tropics compared to the equator, which stands in contrast to the statement of "uniformly upward trends throughout"

page 10, bottom (Figs. 7, 8): not sure these Figures need to be included in the paper, perhaps as supplement is enough? They don't look that much different from the Eulerian versions (as the authors remark) and aren't discussed much either.

page 11, bottom paragraph: this discussion based on differences in how O3 is handled is useful and should be extended a bit: notably, ERA-Interim and MERRA-2 are very different in this regard with ERA-Interim using a climatological O3 product in their radiative scheme and MERRA-2 using its own O3 field – so the effect of O3 on the tropopause and its trends will likely be very different between these two reanalyses.

page 11, line 33: please clarify that you are referring to _anomalous_ upwelling and downwelling (the full residual circulation is still downward over the polar latitudes)

page 12, line 28: awkward sentence structure ("Significant trends . . . were found to be increasing . . .") - please modify

---

## Author Comment (AC1) · 7 Feb 2019

- in page 1 line 23 I miss a citation to Anel et al. (2006). This work also deals with the trends from radiosonde data and indeed it will be useful to discuss some issues later in the paper;

Thank you for the suggestion. The citation has been added.

- in page 2, after line 17: usually there is some confusion on the issue of definition of the tropopause. Words have meanings and being fair it only exists one definition for the tropopause, the one established in 1957 by the WMO. Others are criteria to approach the behavior of the ÂĂÂŹtropopauseÂĂÂŹ or UTLS transition according to the best fit for different studies, campaigns, etc. This does not change the reality of the complex

atmospheric behavior, but using the right words is useful for those not so familiar with the topic that could waste time looking for formal definitions that do not exist anywhere. Therefore in line 18 it is not  ÂŹThe conventional tropopause definitionÂĂÂŹ but ÂĂÂŹThe tropopause definition;

This has been changed to "The *original* tropopause definition" to retain useful context for discussing alternative definitions in the remainder of this paragraph (beginning P2, L13 of the revision).

- in page 2, line 22: in some way linked with the previous issue, I do not think that it is correct to say that there are exceptions to performance. Simply there are regions of the Earth where the UTLS structure is so complex that there is not a tropopause or transition troposphere/stratosphere as such. You mention one case where this behavior is mostly driven by the very specific tropospheric radiative balance during the austral winter. But it is not the only case. The same happens in the third-pole (the Tibetan Plateau) but because of dynamical reasons. There unstable mix of air can make impossible to get a troposphere-stratosphere distinction because of the high altitude of the plateau and its radiative balance (see Chen et al. 2013 and Chen et al. 2016);

Mentions of "performance" were removed and complex, layered stability structures were also acknowledged (P2, L18 of the revision).

- page 3 line 13: indeed fifteen years before Hoinka et al. (1998) had clearly established that the usual values of 1.6 PVU introduced in a campaing in the 1980 ÂŹs or the  ÂŹpopular ÂŹ 2 PVU value underestimate the reality of the tropopause height (obviously in extratropics and polar regions);

The references have been cited, and text has been changed a bit at P3, L13 of the revision.

- subsection 2.1 "Reanalysis output": for the purpose of this work, more relevant than this information (vertical levels and top) is to know the distribution of levels (or dz) between 200 hPa and 50 hPa. I would recommend to the authors to focus the description here on this layer. This will enable them to simplify the understanding and discussion of results later, for example in section 3;

The information of model vertical resolution in the UTLS has been added in Section 2.1, as well as a more direct reference to the Fujiwara et al reanalysis comparison paper.

- page 6 lines 5-10: this is a good exercise to guarantee representation with a case study. But this had already been proved by Antuna et al. (2006) using other station at a quite similar geographical location. I recommend to cite the work to add extra support and to include in the text the coordinates for Corpus Christi (unless I have missed them);

The coordinates for Corpus Christi have been added to the text (P6, L13 of the revision). Citation has not been added because this illustration is dataset specific (i.e., showing the level of detail between full-resolution data and reduced resolution data in IGRA) and the Antuna paper focuses only on mandatory-level radiosondes and the impacts of missing mandatory-level data for climatological analyses.

- in subsection 2.4 you state ÂĂÂŹthe 35-year analysis period. I have not got clearly what is the period of study: 1979-2015? This is 37 years. 1981-2015?. Please, clarify it;

We have added a parenthetical reference to the time period analyzed here (1981-2015) to remind the reader (P6, L28 of the revision).

- page 8, lines 15-16: there is another basis for this (one of them briefly mentioned in the paper), the competing phenomena of tropical widening where the tropical tropopause overlaps the extratropical one and the horizontal meridional entrainment of extratropical air to tropical regions (Wang and Polvani, 2011; Añel et al. 2012; Castanheira and Gimeno, 2011).

Text modified by also acknowledging double tropopause seasonality (P9, L1-3 of the revision).

- subsection 3.2, first paragraph: this is in agreement with the results for the Scenario 1 studied by Añel et al. (2006). That is, raw series without data homogenization. Thought IGRA solved several of the problems that existed in CARDS, here you do not perform any change-point detection technique and this restricts the validity of your results. I think that the issue of not undergoing change-point detection deserves to be mentioned here and that a comparison in the text with the values obtained by Añel et al. (2006) and Santer et al. (2003a,b) would be good as it would enable readers to get a more complete picture of the state-of-the-art.

The point on Siberia deserves special attention in my view: this is also in agreement for with part of the Scenario 1, and with Scenarios 2 and 3 of Anel et al. (2006). Here I would point out two different issues:

1. some of the radiosonde series in this region show up to a 1% significant correlation with the Northern Annular Mode, this could explain partial regional trends. But as soon as in the 1960 ĂŹs Makhover reported that this region has a special behavior in comparison with similar latitudes in this hemisphere (check Antuna et al. 2009 or the original Russian books cited therein);

2. no doubt it deserves a deeper analysis with data homogenization techniques, but there is a potential reason that could explain bias (be aware that I talk about bias not changes in trends) over the region corresponding to the former Soviet Union. This rea- son is the use of different radiosondes with very different equipment than the extended Vaisala RS80/RS90 radiosondes for other parts of the world. A quick check of the metadata in IGRA shows how some stations over the period 1980-1990 there was up to 4 or 5 changes of radiosonde model (changes, not simple updates) and in some of them radiation corrections in 90 ĂŹs. This kind of problems with soundings over Russian territory with frequent radiation corrections was also pointed out by

Makhover (again see Antuña et al. 2009). This could have an impact on any trend computed. Therefore any statement on trends without change point detection and data homogenization should be accompanied of one on the limitations of the data analysis.

Thank you for these comments. A comparison in the text with the values obtained by Anel et al. (2006) and Santer et al. (2003a,b) has been added at P9, L17-19 of the revision. In addition, the limitations of not using change point detection or data homogenization have been acknowledged at P7, L21-25 of the revision.

- subsection 3.3, last sentence: I think that it could exist a partial explanation for this behavior in Fig.4 for CFSR. This is my hypothesis: as it has been proved by Anel et al. (2008) in presence of multiple tropopauses the first lapse rate tropopause (LRT1) is lower than when a single tropopause exist and multiple tropopauses are not present. As Xian and Homeyer show CFSR has lower bias and increased resolution at UTLS levels. This enables this dataset to better represent a bigger number of multiple tropopause events. Having more multiple tropopause events means that an increasing proportion of lower LRT1 cases should be found. This should be more clear in critical regions for the detection, such as subtropics. Therefore the positive trend in the frequency of multiple tropopauses and lower bias of CFSR would be driven an increased frequency of lower LRT1.

Nothing changed. Comparing tropopause altitude trends to the double tropopause trends in CFSR, there is no significant increasing trend in double tropopause frequency in the extratropics where decreasing tropopause altitude was found. Thus, the connection between the decreasing primary tropopause altitudes and increasing double tropopause events mentioned by the referee is not robust in the extratropics for CFSR. Moreover, the remaining reanalyses show increasing double tropopause frequencies and increasing tropopause altitude in most regions.

- page 10, lines 9-15: this is exactly what is stated in Castanheira et al. (2009) (Fig. 8) using IGRA data and a probable consequence of the energetic modes at UTLS levels.

I think that the numbers here obtained should be compared to their ones and the work cited.

Thank you for bringing this work to our attention as we were not aware of this double tropopause trend analysis. We cited the work and compared our results to their trends of double tropopause frequency in two latitude-bands (30-60N and 30-60S) at P11, L11-13 of the revision.

- page 13, line 14: I do not think that "found" is the right word here. To be fair beyond the useful contribution on comparison between state-of-the-art reanalysis, the other results here presented only confirm previous findings existing in the literature and it should be acknowledge in this way.

Replaced "found" with "shown" here (P15, L8 of the revision).

- Table 1: I understand that values in this table are computed using all the stations, independently of the hemisphere. This could provide a sense of average changes, but if you present the results for months representative of seasons, what is the point on mixing NH and SH stations?. Doing such thing does not let to appreciate the true seasonal change. In my view exposing only the values for extratropical regions of one of the hemispheres would be the right way of doing it, as there is no point on including the tropics because of the lack of seasonal variability. Moreover Double tropopauses are a phenomenon with strong seasonal dependence associated to extratropical wintertime UTLS baroclinicity (Castanheira et al. 2009) and therefore the same reasoning applies.

Good points. We have removed the seasonality from Table 1 and shown the total evaluation numbers only. The new Figures 2 and 3 summarize the results for season and location (extratropics, subtropics, and tropics).

---

## Author Comment (AC2) · 7 Feb 2019

**1   Major Comments**

My major comment is that the authors should have included detailed analysis for both bias and RMS while validating. While the authors tend to emphasize the RMS statistics, I hope they know that the RMS, as a loss function, gives a relatively high weight on large errors since it is more sensitive to extreme values on long tails/outliers due to the fact that the errors are squared before they are averaged. So, sometimes very few extreme values can completely change the statistics, which is not desirable for this analysis.

In contrast, the bias is sometimes more intuitive because it tells us how much of abso-

lute differences between the radiosonde and the reanalysis. However, the bias analysis is not perfect because the positive and negative biases will cancel out. I would separate the bias analysis into positive and negative, with positive means the reanalysis primary tropopause showing at a higher altitude, and negative means the reanalysis primary tropopause showing at a lower altitude. Meanwhile, adding the frequency (with respect to total samples considered) of positive and negative bias. In this way, we know that on average how frequent and how much the reanalysis would overestimate/underestimate the tropopause height. I think this detailed analysis is more meaningful to the community.

In this sense, it is fair to always include both bias and RMS error analysis. I think Fig. 2 should include bias on the first panel and RMS error on the second panel. For each panel, different shapes represent different months, but please do include one more statistics for all-season averages. Then, add another figure that repeats a similar analysis for the double tropopause. Having the easy visualization of the statistics, still, keep Table 1 of detailed numbers for easy reference.

Thank you for these comments. We now separate the bias analysis into positive and negative, which is listed in the new Table 1. Bias analysis is also included for primary tropopause altitudes and double tropopause frequency within different latitude bands, with statistics for all-season averages added (new Figures 2 and 3). Necessary changes have been made in Section 3.1 in the revised manuscript.

Another major comment is on the accuracy of IGRA data, and its ability to precisely document the lapse-rate tropopause is crucial for this study. It helps if the authors could iterate in more details on how the $<= 50m$ vertical resolution of raw observation are eventually reported in only 1.5-2.5 km vertical resolution at the UTLS (although in the revised manuscript the authors changed to $> 1km$). The "$> 1$ km" is still less desirable for studying the vertical variability of temperature records - it will miss effects of both gravity waves and the Rossby waves acting on the temperatures. Given the reported resolution, why not using GPS/COSMIC temperature records that has a better

coverage? The focus of the paper is from 5-20 km, in which COSMIC is totally capable of seeing waves on temperatures.

If I understand it correctly, Figs. 1-2 and Table 1 are the only places that the authors performed apple-to-apple comparison by collocating the reanalysis to the radiosonde locations. For all other analysis, the authors just reported trends inferred from gridded results at each latitude-longitude box, so the results could be biased by sampling sizes. So, the first part of validation is more meaningful to my sense. That said, personally I am not interested in the trends reported. For example, what does a trend of +/- 50 m/decade in primary tropopause mean? What does a positive trend of double tropopause frequency mean in specific reanalysis? Unless you can elucidate the possible cause of trends with proof, I don ĂŹt think the trend numbers themselves have significant meaning. On the positive side, the fact that different reanalyses showing different trends is meaningful in that they imply how unreliable the reanalyses are as to the tropopause analysis. This makes me wondering if it is necessary to include the trend analysis, especially in such a large portion of the paper. If I were the authors I would report the bias and errors in more details to help the community to understand the different performances of the reanalyses.

There is perhaps some confusion on the typical resolution of the IGRA profiles here. In the discussion paper, it is stated that reduction of the radiosonde profiles to mandatory and significant levels only *can* result in vertical resolution larger than 1 km, but this is a worst case scenario. As Figure 1 demonstrates, the vertical resolution of the IGRA data is often finer than this. We do not believe additional detail on the process of reporting mandatory and significant levels beyond what is provided in Section 2.2 is necessary, but we have clarified a few points there in the revision.

As for the suggestion to use GPS/COSMIC temperature profiles to investigate tropopause characteristics, we would like to do that in the future. However, an obvious shortcoming of the GPS/COSMIC temperature records is the limited temporal coverage, which is not suitable for studying the long-term changes in tropopause characteristics (the primary of the focus of this paper). We have added acknowledging this on P5, L21 of the revision.

Finally, the purpose of focusing on tropopause trends is motivated by the review given in the Introduction. Namely, tropopause altitude trends are believed to be an indicator of climate change as increases in tropopause altitudes often occur with increases in tropospheric temperatures. In addition, double tropopause occurrences provide a physical perspective of UTLS dynamics (most notably STE). Evaluating long-term trends provides a unique insight into these processes. Comparing model trends with observed trends are also another method of model validation, so we believe analysis of trends is well-justified and relevant to the scientific community (as also evidenced by the remaining reviews of the paper).

As for elucidating the sources of the trends, we have drawn on complementary results from other recent efforts in Section 4. We have also expanded some of the discussion on these trends there by considering potential physical and dynamical sources (most on P13 of the revision).

A last comment is that I do hope the authors could put more emphasis on the physical meaning/causes of the (large) differences among different reanalysis. So, beyond the vertical resolution, could there be any other reasons that caused the discrepancies? The current version seems to be less scientific and more like a technical report.

We have added text and additional analysis evaluating the effects of vertical resolution (see Tables 1 and Figures 2 & 3 of the revision). Apart from vertical resolution, it is quite difficult and beyond the scope of this study to identify the reasons for oftentimes subtle differences between the tropopause trends in the reanalyses. The tropopause reflects the combined impacts of a long list of choices in model design and assimilated data, so elucidating the role of each in controlling trends in tropopause characteristics is a daunting task. Vertical grid spacing is an ideal target for initial evaluation, given that the tropopause definition depends greatly on it. Thus, we have limited our detailed

evaluation to this single source in the revision. To examine the role of other aspects of the model design, comprehensive sensitivity studies using a single modeling system are likely required.

**2 Minor wording comments**

1. P1L8, attributed $>$ attributable

Corrected.

2. P1L9: observations

Done.

3. P1L9: and reanalyses $>$ and the reanalyses

Done.

4. P1L9: analysis period

analysis period has been clarified.

5. P2L10: $>$ the UTLS composition

Not changed.

6. P1L13-15: this sentence doesn'ÂĂÂŹt make sense

Revised to improve clarity (P1, L12-14 of the revision).

7. P3L6-7: this makes sense because of the existence of the ozone layer, but can you be more specific about it?

This point has been clarified (P3, L4-8 of the revision).

8. P4L1: I donâÂĂÂŹt understand the logic here. If the radiosonde data is so limited, why bothering using them instead of COSMIC data? Plus, this part sounds like belonging to the discussion part.

See previous response to similar comments.

9. P5L10-12: all reanalyses are reported in sigma or eta coordinates. From conversion you might get temperatures on pressure levels easily, but how did you get them in altitude coordinates? Did you use simultaneous geopotential heights to interpolate the data? Be more specific about how you preprocessed the data.

This point has been clarified at P5, L12 of the revision.

10. P5L16: > quality-controlled

Corrected.

11. P5L23-24: one comment is that this linear interpolation doesn't change the shape of the profile, at all. So, you typically end up with the same value without doing interpolation.

An interpolation is needed to verify the second criterion of the WMO definition and the criterion for identifying multiple tropopauses. We have clarified the value provided by interpolation in Section 2.2.

12. P7L18: tropopause altitudes in MERRA-2 > primary tropopause altitudes in MERRA- 2.

Done.

13. P8L16: is maximum > its maximum

Done.

14. P13L11-12: how did you reach this conclusion?

Similar to Rossby wave breaking leading to transport of tropical UT air into the extratropical LS, double tropopauses can be formed by equatorward transport of extratropical LS air into the tropical UT during wave breaking events. We have added a relevant citation to this conclusion [Liu and Barnes (2018)].

References:

Liu, C., and Barnes, E. A. (2018). Synoptic formation of double tropopauses. Journal of Geophysical Research: Atmospheres, 123, 693–ÂĂÂŞ707.

---

## Author Comment (AC3) · 7 Feb 2019

**1   Main aspects:**

1) Concerning the reproducibility of results, the paper lacks information about the radiosonde data used in the study: how were IGRA stations selected in the first place? The number of selected stations (317) and the corresponding amount of observations for 1985-2015 are given later in the results section, with their approximate locations shown in the Figures.  But IGRA (version 2 released in 2016) contains temperature data from 800-ÂÃÂŞ900 radiosonde stations within the studied period. Nothing, however, is said about the choice of stations, concerning the homogeneity of time-series in terms of temporal and vertical features (i.e., leaving aside the much more difficult

problem of instrument biases): temporal regularity and continuity; vertical resolution around the tropopause.

We selected the radiosonde observations based on both complete vertical profiles and the homogeneity of time-series, as we had previously outlined in Sections 2.2 and 2.4. We have added a few clarifications to these sections to emphasize some key points related to analysis of the radiosonde data.

2) A linear interpolation to a 200-m regular vertical grid was applied prior both to radiosonde and reanalysis temperature data before tropopause identification. The authors claim this was done ÂĂÂIJin order to enable reliable tropopause identification. This phrase is potentially confusing to the reader. Evidently, an interpolation is needed to verify the second condition of WMOâÂĂÂŹs definition of first tropopause, as well as to look for a second tropopause. But a linear interpolation simply does not change the lapse rate between the known data points. So, the estimation of the first and second tropopause levels is essentially limited by the resolution of data âÂĂÂŞ as the authors in fact recognize in other parts of the paper. The gain resulting from the interpolation scheme should be explained to make this point clear.

Thank you for identifying an opportunity to improve clarity. As correctly inferred, the value gained from linearly interpolating the radiosonde data to a higher-resolution regular grid spacing is to enable thorough evaluation of the second WMO criterion and the criterion for identifying multiple tropopauses. We have clarified these points in Section 2.2.

3) Radiosonde data were analyzed at the principal synoptic hours, 0000UT and 1200UT, whereas reanalysis data were analyzed only at 0000UT. This means that half of the time-zones on the global reanalysis fields of temperature (at latitudes outside of the polar regions, after averaging over one or more years) is represented by daylight times, while the other half is represented by nocturnal times. In this respect, in Figs. 3-6 it is not clear why some radiosonde stations show 0000UT average values while

others show 0012UT values, since reanalysis-derived values refer always to 0000UT. Also, considering the diurnal variations of the tropopause height, it should be explained how the radiosonde-reanalysis tropopause differences listed in Table 1 were exactly calculated.

Although there is a diurnal cycle of tropopause height, the long-term tropopause trends from the reanalyses at different synoptic times are consistent (not shown). The comparisons listed in Table 1 are based on 00 UTC profiles only. This point has been clarified in Section 3.1.

4) Although not obligatory, to be more informative Table 1 should depict hemispheric seasons. Or perhaps individual months, but then restricting to North Hemisphere, where the amount of radiosonde data (used as reference to errors) is much larger there than in the South Hemisphere.

Rather than restricting values in the table to North Hemisphere only, the new Figures 2 & 3 satisfy this suggestion.

5) The calculation of tropopause altitude needs a bit of clarification: is moisture included in the hypsometric equation? Tropopause altitude refers to geometric altitude or geopotential altitude?

Before tropopause identification, geopotential height was computed for each reanalysis model-level output using the moisture-included hypsometric equation. Therefore, tropopause altitude refers to the geopotential altitude. This has been clarified in Section 2.3.

6) Maybe the large discrepancies between the results obtained from CFSR and the other reanalysis models (seen in all plots) deserve a slight explanation.

We have expanded discussion of trends and their potential ties to physics/dynamics in the Conclusions and discussion section. Some additional analysis was included, but the source of the discrepancies in CFSR relative to the remaining reanalyses remains

unclear.

**2 Secondary aspects:**

P2, L19. Where it reads âÂĂÂİJ(. . .) (also known as vertical temperature gradient) (. . .) âÂĂÂİ it correctly should read âÂĂÂİJ(. . .) (negative of the vertical temperature gradient) (. . .)âÂĂÂİ

Corrected.

P4, L1. âÂĂÂİJ(. . .) since they are only launched from land massesÂĂÂİ. Considering the radiosondes launched on whether ships and âÂĂŸships of opportunity (even if not used in the study) it should be better to write âÂĂÂİJ(. . .) since they are mostly launched from land-masses.

Good point. Corrected.

P4, L6. âÂĂÂİJReanalyses assimilate global high-quality observations (. . .). Do not forget to mention other observation platforms besides radiosondes. Moreover, I doubt that all observations assimilated in reanalysis models are of high-quality. A meteorological reanalysis is supposed to deal with inaccurate and incomplete observations to some degree. ÂĂÂİJQuality-controlledÂĂÂİ is closer to reality.

Replaced with "quality-controlled".

P4, L 30. I don't understand the words âÂĂÂİJa physical perspective of the UTLS. I suppose that the authors point is that their paper provides an evaluation of reanalysis-model performance regarding the UTLS temperature structure.

Since there is a close correlation between double tropopause occurrence and STE events and the tropopause is a physical attribute of the atmosphere, tropopauses can be used to diagnose UTLS dynamics. The use of the term "behavior" seems to have

been the source of confusion here, so we've replaced it with "dynamics" (P4, L31 of the revision).

P6, L12. Thus, we are confident that IGRA data are suitable for tropopause analyses following the methods employed here.ÂĂÂÌ How can you tell, from a demonstration with two random soundings from one site? The study uses nearly $10^5$ soundings from over 300 radiosonde stations! The above assertion is not acceptable. Although Fig. 1 serves the purpose of illustration of the idea, paradoxically, expressing here some uncertainty would give more confidence to the reader.

Excellent point. We have added a few clarifying bits of information here to address this issue. We did not limit this type of evaluation to a single station and did randomly select from alternative locations and time periods where we had access to the full resolution data. The point of this comparison is to demonstrate that mandatory and significant levels are sufficient for tropopause identification. We have acknowledged that results for alternative locations and times are consistent with that shown here and that differences in tropopause identifications between full resolution and reduced resolution profiles are ≤100 m (P6, L16 of the revision).

P7, L14-16. It'ÂĂÂŹs not totally clear whether Fig. 3 (and so on) uses only four months per year or not.

It is stated throughout the paper that trend analyses are based on monthly mean fields. Since this was not a common source of confusion for the reviewers, we have decided that additional clarification is unnecessary.

P13, L16. ÂĂÂIJ(. . .) increases in primary tropopause altitude are associated with a warming climate (. . .). The suggested connection is supported by a very few modeling experiments until now. I'd replace "ÂĂÂIJare" by something less assertive like "ÂĂÂIJis probably" or "ÂĂÂIJis believed to be".

Replaced by "is believed to be".

Fig. 6 and Fig. 8. If possible, the color scale legend "ÂĂÂIJDouble tropopause fre-quency" should be changed to "ÂĂÂIJDouble tropopause trendÂĂÂì".

These legends have been changed to "Double Tropopause Frequency Trend".

---

## Author Comment (AC4) · 7 Feb 2019

**1   General comment**

1) The fairly large bias in MERRA-2 is interesting and I was surprised that it didn'ÂĂÂŹt receive more attention by the authors (at least not in the writeup). After all, this is a (re)analysis, i.e., it includes a modern data assimilation scheme, presumably assimilating the radiosonde observations that here used as a reference. So my expectation was that all modern reanalyses essentially reproduce the tropopause. Fig. 1 furthermore stimulates suspicion: how can a reanalysis have such large temperature biases ($>$ 5 K!!) in the upper troposphere? Without labelling I would have guessed that this is a free-running model. Don't you expect all modern reanalyses to very closely agree

about temperature in the upper troposphere? This is the case between the other three products: ERA-Interim, JRA-55, CFSR. Is this simply an outlier example or do you often find such large biases in MERRA-2? Is this something that's documented in the literature? To be honest, if this is a robust bias in MERRA-2, then this product shouldn'ÂĂÂŹt be used for UTLS studies . . . in any case, this requires more discussion by the authors.

After careful re-evaluation of the MERRA-2 fields we were using for the profiles in Figure 1 (added quickly after the request during initial review before passed on to open discussion), we discovered that the wrong reference levels were used. Instead of using the pressures and altitudes in the middle of the model layers that correspond to the temperatures, we were using the pressures and altitudes of the model levels (the edges of the layers). This resulted in an artificial displacement of the profile of approximately 500 m in the UTLS. We have corrected this error in Figure 1 and the remaining analyses in the paper, for which it had little impact on the results (except for the bias analysis). The revised analyses clearly show that MERRA-2 is consistent with the remaining reanalyses in its representation of UTLS temperatures. Many thanks to the reviewer for emphasizing this point.

2) Vertical resolution is mentioned at many places to potentially explain differences between radiosondes and reanalyses. Isn'ÂĂÂŹt this easily testable? You could degrade the radiosondes to the model resolutions and see if that really explains the differences. You could even study some of the characteristics (e.g., double tropopause frequency) as a function of vertical resolution by gradually degrading the radiosonde data. Perhaps the authors have already tried this, in any case, I would strongly suggest to include corresponding results / discussion in the paper.

Thank you for the suggestion. We have degraded the radiosonde observations to the vertical grid of each model and recomputed the bias and RMS differences. Bias and RMS differences in instantaneous primary tropopause altitudes show little sensitivity, but large reductions in both are found for double tropopause frequencies. This point

has been clarified in Sections 3.1 and 4 in the revised manuscript and reflected in the revised analysis presented in new Figures 2 & 3 and Table 1.

**2   Minor comments**

page 2, line 16: "uncertainty that is comparable to the vertical resolution of the model ÂĂÂÎ"ÂĞ this makes intuitive sense, but is this a priori clear given that you interpolate between levels for the tropopause calculation?

The value given by interpolation of the temperature profiles has been clarified in Section 2.2. The interpolation only assists in routinely satisfying the second criterion of the WMO definition and the criterion for identifying multiple tropopauses. Therfore, yes, we do expect it to be clear a priori that uncertainty should be comparable to the vertical resolution of each model.

page 2, line 19: the lapse rate is equal to minus the vertical temperature gradient

Corrected.

page 2, line 28-29: Anel ref'ÂĂÂŹs

Reference has been added at P2, L25-27 of the revision.

page 3, line 7-8: sentence doesn't work like this; how about: "PV, which is conserved . . ., is commonly used for transport studies in the extratropics and often used to define a dynamical tropopause . . ."

Done.

page 3, line 10: "threshold used varies considerably"ÂĘ seems like an exaggeration (I'd suggest to remove ÂĂÂIJ"considerably"ÂĂÂÎ), note a lot of the STE studies (e.g., Wernli group and others) use 2 PVU and this value seems to be used mostly

Done.

page 5, lines 11-12: these are somewhat subjective choicesÂ̧ have you checked the corresponding sensitivity? E.g., are the results sensitive to obtaining tropopause levels from the native horizontal and vertical grid, and interpolating to the 1-by-1 lon-lat grid afterwards? IâÂ̆ÂŹm also not sure I understand the purpose of oversampling to the 200-m grid in the vertical for tropopause identificationÂ̆ please provide rationale (relevant for line 24 as well).

We have evaluated the sensitivity to these choices and it is negligible. Interpolation in the horizontal dimension has no effect on the tropopause other than reducing the level of horizontal detail (which is advantageous for apples-to-apples comparisons of the reanalyses and is how we have locally archived the data for long-term use). Some text has been added to reflect the lack of sensitivity to the choice of synoptic time here (Section 2.1). In addition, see previous response for detail on the need for vertical interpolation prior to tropopause identification.

page 6, line 24: how do you assess whether data points are roughly evenly distributed?

We checked the length of time gaps in the tropopause altitude time series for each station, and selected the stations with maximum gap duration less than 5 years (there were only 59 stations included with gaps longer than 3 years and these were manually evaluated to confirm there were no deleterious effects on the trend analysis - e.g., missing long time chunks at the beginning and end of the 35-year analysis period). This point has been clarified at P2, L33 of the revision.

page 7, line 8-9: do you do this separately for the two hemispheres? How do you then handle the equator, which in the relative coordinates "Â̆ÂİJmoves"Â̆Âİ around?

Yes. For plotting, any data extending beyond the equator is trimmed. We have added some clarifying points in Section 2.4.

page 7, line 13-14: so here you suggest that you do use the native model grids for

tropopause calculations, in contrast to the description on page 5 please clarify

This point has been clarified at P7, L30-31 of the revision.

page 7, line 18: is this bias a function of latitude?

This bias is derived from global observation, the variation with latitude has been included in new Figures 2 and 3.

page 8, discussion of Fig. 2: have you considered normalizing these RMS differences by a measure of internal/natural variability (e.g., interannual standard deviation)? Larger RMS differences would be expected in regions with larger internal variability, so part of the latitudinal differences could be related to different internal variability.

Indeed, the large internal variability can result in large RMS error in some regions, such as the extratropics. The variability of tropopause altitude in these regions is mainly attributed to the subtropical jet shifting latitude, which is associated with north-south migration of the tropopause break. We have not attempted to normalize these RMS differences in the revision, but have expanded the bias analysis to reflect points raised by other reviewers.

page 9, line 5: over the Atlantic trends are larger at the edges of the tropics compared to the equator, which stands in contrast to the statement of "uniformly upward trends throughout"

This has been changed to "larger upward trends".

page 10, bottom (Figs. 7, 8): not sure these Figures need to be included in the paper, perhaps as supplement is enough? They don'ÂĂÂŹt look that much different from the Eulerian versions (as the authors remark) and aren'ÂĂÂŹt discussed much either.

We believe the differences between these tropopause break-relative analyses and the Eulerian analyses, though small in some respects, are important to show in the paper and to the discussion included (despite the fact that it is relatively brief).

page 11, bottom paragraph: this discussion based on differences in how O3 is handled is useful and should be extended a bit: notably, ERA-Interim and MERRA-2 are very different in this regard with ERA-Interim using a climatological O3 product in their radiative scheme and MERRA-2 using its own O3 field so the effect of O3 on the tropopause and its trends will likely be very different between these two reanalyses.

The description of differences in ozone assimilation between reanalyses has been expanded beginning at P13, L34 of the revision.

page 11, line 33: please clarify that you are referring to anomalous upwelling and downwelling (the full residual circulation is still downward over the polar latitudes)

Corrected.

page 12, line 28: awkward sentence structure ( ÂĹJSignificant trends . . . were found to be increasing . . . Âİ) - please modify

This has been changed to "Significant increasing trends in double tropopause frequency were found nearly everywhere in the radiosonde observations ...".

---

## Author Response (AR2)

Editor comments in black, author responses in blue.

1. Fill in the copyright statement in page 1
   The authors did not intend to include a copyright statement on page 1, so this has been removed.

2. I found a spelling error on page 9, line 3 (tropoapuses)
   Thank you. This has been corrected.

3. Consider the number of significant places in your table. Three places seem to be a bit much.
   The number of significant places has been reduced from 3 to 2.

[revised manuscript text omitted]